# Understanding hydrologic controls of sloping soil response to precipitation through Machine Learning analysis applied to synthetic data

Daniel Camilo Roman Quintero[1], Pasquale Marino[1], Giovanni Francesco Santonastaso[1], Roberto Greco[1]

[1]Dipartimento di ingegneria, Università degli Studi della Campania 'Luigi Vanvitelli', via Roma 9, 81031 Aversa (CE), Italy;

*Correspondence to*: Daniel Camilo Roman Quintero
(danielcamilo.romanquintero@unicampania.it)

**Abstract:**

Soil and underground conditions prior to the initiation of rainfall events control the hydrological processes that occur in slopes, affecting the water exchange through their boundaries. The present study aims at identifying suitable variables to be monitored to predict the response of sloping soil to precipitation. The case of a pyroclastic coarse-grained soil mantle overlaying a karstic bedrock in the Southern Apennines (Italy) is described. Field monitoring of stream level recordings, meteorological variables, and soil water content and suction has been carried out for few years. To enrich the field dataset, a synthetic series of 1000 years has been generated with a physically based model coupled to a stochastic rainfall model. Machine Learning techniques have been used to unwrap the non-linear cause-effect relationships linking the variables. The k-means clustering technique has been used for the identification of seasonally recurrent slope conditions, in terms of soil moisture and groundwater level, and the Random Forest technique has been used to assess how the conditions at the onset of rainfall controlled the attitude of the soil mantle to retain much of the infiltrating rainwater. The results show that the response in terms of the fraction of rainwater remaining stored in the soil mantle at the end of rainfall events is controlled by

soil moisture and groundwater level prior to the rainfall initiation, giving evidence of the activation of effective drainage processes.

**Keywords:** Water storage, slope response, underground antecedent conditions, hydrological controls, Random Forest, k-means clustering

## 1. Introduction

Slope response to precipitation is highly non-linear, in terms of runoff generation, rainwater infiltration and subsurface drainage processes, which are mostly depending on the initial soil moisture state at the onset of each rainfall event (Tromp-Van Meerveld and McDonnell, 2006b; Nieber and Sidle, 2010; Damiano et al., 2017). The initial (or antecedent) conditions are related to hydrological processes that occur in the slopes, which control how they exchange water with the surrounding systems (i.e., atmosphere, surface water, deep groundwater). These processes occur through the boundaries of the slope, and often evolve over timescales of weeks or even months, much longer than the duration of rainfall events, typically ranging between some hours and few days.

While the importance of soil moisture conditions on slope runoff and drainage has been recognized long since (Ponce and Hawkins, 1996; Tromp-Van Meerveld and McDonnell, 2006a, 2006b), only recently the scientific community started providing new perspectives to better understand hydrologic conditions predisposing slopes to landslides (Bogaard and Greco, 2018; Greco et al., 2023), to explain why most of large rain events do not destabilize slopes, while only some do (Bogaard and Greco, 2016), and physically based models capable of integrating hydrological knowledge for predicting landslide occurrence have been proposed (e.g., Bordoni et al., 2015; Greco et al., 2018; Marino et al., 2021).

The triggering of some rainfall-induced geohazards, such as shallow landslides and debris flows, is favoured by pore pressure increase, caused by rainwater infiltration and consequent soil moisture accumulation. The storage of rainwater

within the soil requires drainage mechanisms developing in the slopes in response to precipitation to be not so effective to drain out much of the infiltrating water (Greco et al., 2021; 2023). Consequently, especially for nowcasting and early warning purposes, the identification of hydrological variables suitable to identify slope predisposing conditions is extremely useful. Thus, to better understand how hydrological predisposing conditions may control the processes involving the sloping soil response in terms of water storage, field monitoring for the assessment of the slope water balance is highly recommended (Bogaard and Greco, 2018; Marino et al., 2020a).

The identification of suitable variables to be monitored in the field is indeed useful to achieve an insight of the behaviour of the interconnected hydrological systems (i.e., groundwater, surface water, soil water). Besides the study ofrainfall-induced landslides, the evaluation of the hydrological scenarios in a region of interest could impact several other applications, from flood hazard assessment (Reichenbach et al., 1998; Forestieri et al., 2016; Chitu et al., 2017), to the prediction of possible crop water stress conditions in relation to defoliation (Capretti and Battisti, 2007), pathogen expansions in chestnut grove (Gao and Shain, 1995), and plant mortality in a climate change context (McDowell et al., 2008).

This research focuses on a case study of a slope located in Campania (southern Italy), representative of a wide area frequently hit by destructive rainfall-triggered shallow landslides (e.g., Fiorillo et al., 2001; Revellino et al., 2013). In fact, such geohazards are recurrent along the carbonate slopes covered with unsaturated air-fall pyroclastic deposits, diffuse over an area of few thousand square kilometres around the two major volcanic complexes of the region, the Somma-Vesuvius and the Phlaegrean Fields (Di Crescenzo and Santo, 2005; Cascini et al., 2008). The underlying limestone bedrock, densely fractured, is characterised by the presence of deep karst aquifers (Allocca et al., 2014). The

triggering mechanism of landslides in the area is the increase of water storage within the soil mantle after intense and persistent precipitation, leading to pore pressure build up (Bogaard and Greco 2016). Slope equilibrium is in fact guaranteed by the additional shear strength promoted by soil suction (Lu and Likos 2006; Greco and Gargano 2015), which reduction often leads to slope failure due to shear strength loss by soil wetting during rainwater infiltration (Olivares and Picarelli, 2003; Damiano and Olivares, 2010; Pagano et al., 2010; Pirone et al., 2015).

Recent studies show that the response of the soil mantle to precipitation in the study area is affected not only by rainfall characteristics and antecedent soil moisture, but also by the wetness of the interface with the underlying bedrock, which controls the leakage of water into the underlying fractured limestone (Marino et al., 2020a; 2021). At the contact between soil and bedrock, intense weathering modifies the physical properties of the soil as well as of the fractured bedrock, which form a hydraulically interconnected system, the epikarst (e.g., Perrin et al., 2003; Hartmann et al., 2014; Dal Soglio et al., 2020). The changing hydraulic behaviour of the soil-bedrock interface can be related to the storage of water in the epikarst, where a perched aquifer forms during the rainy season (Greco et al., 2014, 2018).

The aim of this study is to identify the major hydrological processes controlling the response to precipitation of the pyroclastic soil mantles typical of the area, and the seasonally recurrent conditions that affect their attitude to retain much of the infiltrating rainwater, through suitable measurable variables. To this aim, a rich dataset of measured rainfall events and corresponding hydrological effects would be required, which was not available for the case study, where monitoring activities had been carried out for few years. Therefore, a synthetic 1000 years hourly dataset was generated, by means of a stochastic rainfall model and a simplified physically based model of the slope, coupling the unsaturated

pyroclastic soil mantle and the underlying perched aquifer (Greco et al., 2018).
Both models had been previously calibrated and validated on field experimental
data (Damiano et al, 2012; Greco et al., 2013; Comegna et al., 2016; Marino et
al., 2021). The synthetic data of soil suction, water content and aquifer water
level, all measurable in the field and assumed as representative of real conditions,
were analysed as if they were measured data. After sorting the rainfall events
within the 1000 years timeseries, a dataset was built with the antecedent
conditions one hour before the beginning of each rainfall event. It included the
previously listed variables plus the total event rainfall depth, and the change in
the water stored in the soil mantle at the end of each rainfall event. To disentangle
the non-linear processes controlling the hydraulic behaviour of the slope, and
their role on the soil response to precipitation, the dataset was analysed with
Machine Learning (ML) techniques, i.e., clustering, and random forest. Indeed,
ML allows managing big amounts of data, such as those provided by assimilation
of extensive monitoring networks, remote sensing, satellite products and other
sources, without introducing any mathematical model structure to highlight the
cause-effect relationships linking the variables.

## 2. Materials and methods

The studied slope, described in section 2.1, belongs to the Partenio Massif, and it has the typical characteristics of many pyroclastic slopes of Campania (southern Italy) (Greco et al., 2018). Indeed, three major zones characterized by unsaturated pyroclastic deposits can be identified in Campania (Cascini et al., 2008): Campanian Apennine chain, composed by carbonate rock covered by a variable layer of pyroclastic soil (from 0.1 to 5 m); Phlegraean district, formed by underlying densely fractured volcanic tuff bedrock, placed under several meters of pyroclastic soils; and Sarno and Picentini Mountains, where a thin layer of pyroclastic material is over a terrigenous bedrock. In these three areas, the thickness of the soil mantle is quite variable, according to the slope inclination and to the distance from the eruptive centre (De Vita et al., 2006; Tufano et al., 2021).

To identify the seasonally recurrent conditions that affect the attitude of the soil mantle to retain much of the infiltrating water, a large set of measurements of rainfall events, and their effects on the slope, would be required. Hence, to enrich the data available from the monitoring activities carried out for some years at the slope (Marino et al., 2020a), a synthetic dataset of the hydrologic response of the slope to precipitation, has been generated with a NSRP stochastic model of rainfall (Rodriguez-Iturbe et al., 1987) and a simplified 1D model of the interaction of the unsaturated pyroclastic soil mantle with the underlying perched aquifer forming in the epikarst. Both the models, described in the following sections, had been previously developed based on experimental data (Greco et al., 2013; 2018; Marino et al., 2021). The obtained synthetic dataset has been compared to the limited dataset from field monitoring, showing a reasonable agreement. Therefore, it has been considered suitable to reproduce slope response to climate forcing, in terms of soil volumetric water content and perched aquifer water level, in the studied area (see Section 2.2).

The synthetic dataset has been analysed with Machine Learning techniques
(Section 2.3), as they result quite powerful to identify non-linear cause-effect
relationships between variables, without introducing any model structure, as if
the data were provided by field measurements. Figure 1 shows the flowchart of
the entire methodology.

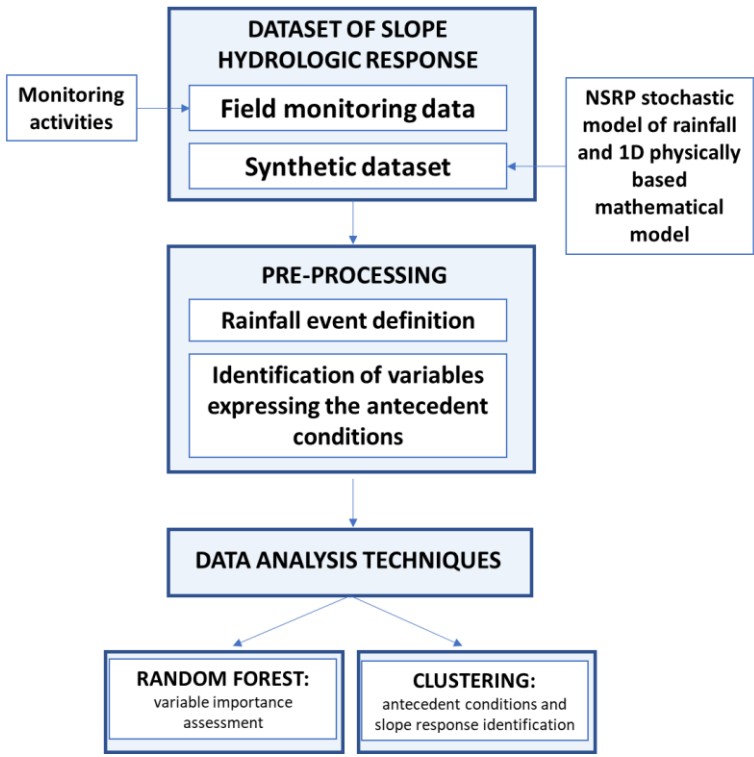


**Figure 1. Flowchart summarizing the methodology followed in the analysis of sloping soil response to precipitation.**

### 2.1.    Case study

The study area refers to the north-east slope of Monte Cornito, part of the Partenio
Massif (Campania, southern Italy), 2 km from the town of Cervinara, about 40
km northeast of the city of Naples. The slope was involved in a series of rapid
shallow landslides after a rainfall event of 325 mm in 48 hours during the night
between 15–16 December 1999, causing casualties and heavy damages (Fiorillo
et al., 2001). A field monitoring station was installed nearby the big landslide
scarp since 2001. Further details of the investigated zone, with indications of the
area affected by the largest of the landslides triggered in 1999, are shown in
Figure 2.

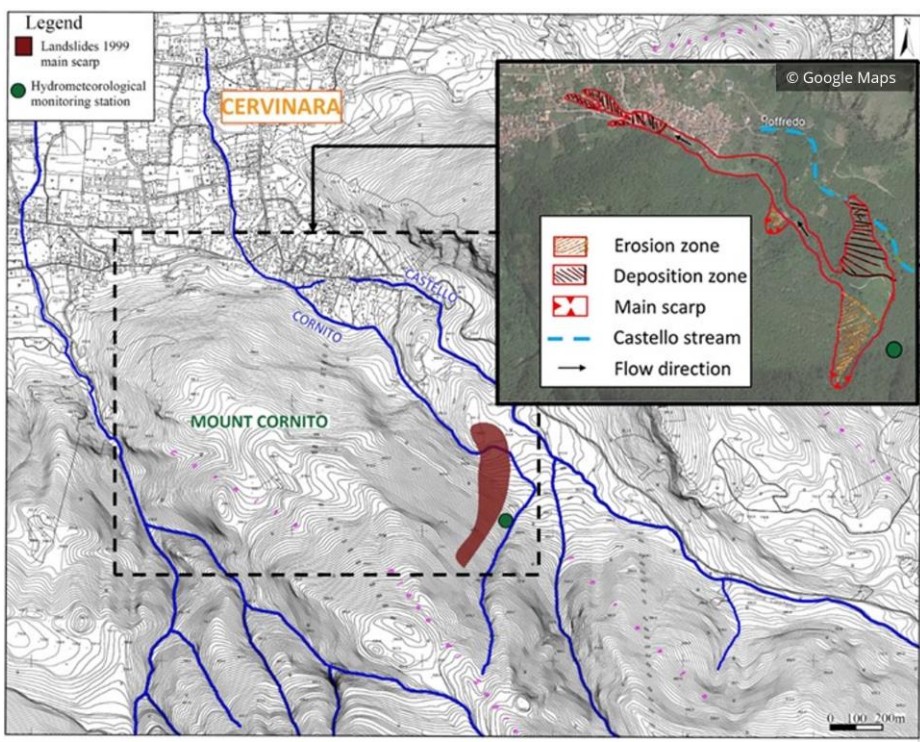


**Figure 2. Location of the study area and indication of the zone affected by a large**
**landslide in 1999. Adapted from: Marino et al. (2020a).**

Partenio Massif is part of the southern Apennines area. The bedrock mainly
consists of Mesozoic-Cenozoic fractured limestones, mantled by loose
pyroclastic deposits, resulting from the explosive volcanic activity of Somma-
Vesuvius and Phlegrean Fields, which occurred over the last 40.000 years
(Rolandi et al., 2003).
The fractured limestone formations of the southern Apennines often host large
karst aquifers, through which a basal groundwater circulation occurs, for which
regional groundwater recharge between 100 and 500 mm/year has been
estimated, with 200 mm/year regarding the area of Cervinara (Allocca et al.,
2014). Moreover, recent studies showed that, in the upper part of the karst system,
denoted as epikarst (Hartmann et al., 2014), more permeable and porous than the
underlying rock, a perched aquifer often develops (Williams, 2008; Celico et al.,
2010). It temporally stores water and favors the recharge of the deep aquifer
through the larger fracture system. The water, which is accumulated temporally
in the epikarst, also reappears at the surface in small ephemeral streams.
Specifically, the slope of Cervinara has an inclination between 35° and 50°, at an
elevation between 500 m and 1200 m above sea level. The soil mantle, usually
in unsaturated conditions, is the result of the air-fall deposition of the materials
from several eruptions, so it is generally layered. It mainly consists of layers of
volcanic ashes (with particle size in the range of sands to loamy sands) alternating
with pumices (sandy gravels), laying upon the densely fractured limestone
bedrock. Near the soil-bedrock interface, a layer of weathered ashes,
characterized by finer texture (silty sand), with lower hydraulic conductivity,
moderate plasticity and low cohesion, is often observed (Damiano et al., 2012).
The soil mantle thickness varies spatially from a minimum of 1.0 m, in the
steepest part of the slope, to larger values at its foot (up to 4-5 meters). The thin
soil mantle, compared to the slope width and length of hundreds of meters (Figure
2), makes the flow processes nearly one-dimensional, except for the close
proximity to geometric singularities.
The pyroclastic soils of the profile are characterized by high porosity (from about
50% for the pumices, to 75% for the ashes) and quite high values of saturated
hydraulic conductivity (ranging up to the order of $10^{-5}$ m/s). Thus, this kind of
soil lets rainwater infiltrate even during the most intense rainfall events, with little
runoff generation, and it can store a large amount of water without approaching
saturation. The values of soil capillary potential, measured during the rainy

season, rarely exceed -0.5 m, as observed also in other slopes of the area (Cascini et al., 2014; Comegna et al., 2016; Napolitano et al., 2016).

The climate is Mediterranean, which is characterized by dry and warm summer and rainy autumn and winter, with mean annual precipitation of about 1600 mm, mostly occurring between October and April. The total potential evapotranspiration $ET_0$, estimated with the Thornthwaite formula (Shuttleworth, 1993), is between 700 mm and 800 mm in the altitude range between 750 m and 400 m (Greco et al., 2018). The vegetation mainly consists of widespread deciduous chestnuts, with a dense understory of brushes and ferns, growing during the flourishing period (between May and September). In fact, visual inspections of the soil profile showed a large amount of organic matter and roots. In most cases, roots are denser in the uppermost part of the soil mantle and become sparse between the depth of 1.50 m and 2.00 m below the ground surface, reaching the basal limestones and penetrating the fractures.

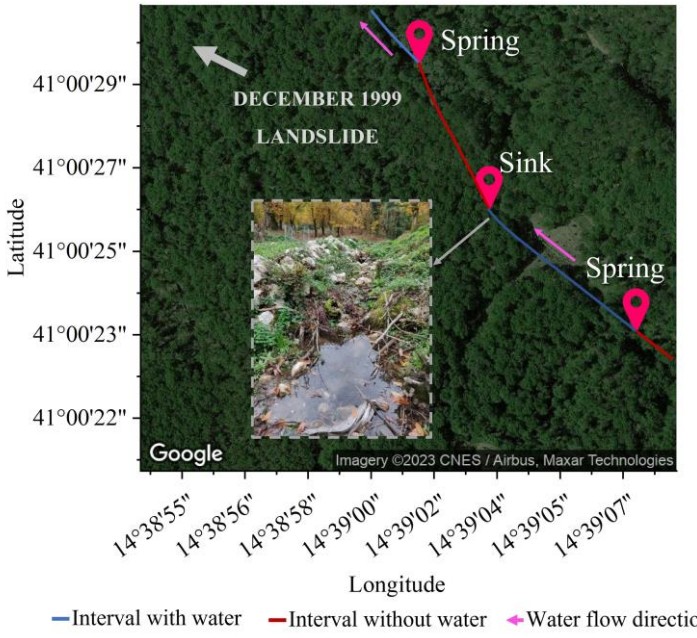

**Figure 3. Identification of surface water flow in the Castello stream at the beginning of the rainy season in November 2021 by visual recognition of springs and sinks in the watercourse**

Moreover, in the surrounding area, several ephemeral and perennial springs are
present, mostly located at the foot of the slopes, which supply a network of small
creeks and streams, allowing to show the activity of the aquifer discharge to the
surface water. An indication regarding the Castello stream (the main stream for
this side of the basin), with springs, is shown in Figure 3, where, during a field
recognition in November 11[th] 2021, the surface water flow appeared (springs)
and disappeared (sinks) in some points along the stream course. Normally the
stream exhibits its lowest water depth values up to the beginning of the late
autumn (Marino et al., 2020a, p.3.3), but it is interesting to note that the surface
water in the stream emerging from the epikarstic springs is an indicator of the
active slope drainage.

### 2.1.1.  Field monitoring data

Several hydrological monitoring activities have been carried out at the slope of
Cervinara since 2001, initially consisting of measurements of precipitation and
manual readings (every two weeks) of soil suction by "Jet-fill" tensiometers,
equipped with a Bourdon manometer (Damiano et al., 2012). Afterwards, since
November 2009, an automatic monitoring station has been set at an elevation of
585 m a.s.l., near a narrow track close to the landslide scarp of December 1999.
The installed instrumentation consisted of tensiometers, time domain
reflectometry (TDR) probes for water content measurements, and a rain gauge
(Greco et al., 2013; Comegna et al., 2016).
Since 2017, the hydro-meteorological monitoring was enriched (Marino et al.,
2020a), aiming at understanding the seasonal behaviour of the slope and the
interactions between the hydrological systems, i.e., the unsaturated soil mantle,
the epikarst, and the underlying fractured bedrock.
Specifically, the data collected by tensiometers and TDR probes were
supplemented with those from a meteorological station (composed by a thermo-

hygrometer, a pyranometer, an anemometer, a thermocouple for soil temperature measurement, and a rain gauge), and with the water level in two streams at slope foot, so to gain useful information for the assessment of the water balance of the studied slope.

The data from field monitoring, carried out between 2017 and 2020 with hourly resolution, consist of rainfall, evapotranspiration, soil moisture and suction at various depths, and the water depth of the Castello stream. The data have been useful to highlight seasonally recurrent soil moisture distributions. More details about the measured data and the observed recurrent seasonal behaviour of the area of Cervinara can be found in Marino et al. (2020a).

## 2.2. Synthetic dataset

Aiming at identifying suitable variables to be monitored in the field for the identification of the conditions controlling different slope responses to the precipitation, a rich dataset of rainfall and underground monitored variables, such as soil moisture and groundwater level, is needed. However, a complete field monitored dataset is not always possible to be analyzed and, when it exists, it is commonly available for short periods, granting a relatively small number of measurements. Hence, a synthetic dataset, aiming at improving the information obtained from field monitoring, has been generated. This dataset has been obtained by means of the physically based mathematical model described hereinafter (section 2.2.2). The model has been run with a 1000 years synthetic hourly rainfall series, obtained with a stochastic rainfall generator, for which further details are given in section 2.2.1. The choice of such a long synthetic series has been made to obtain an amount of data, representative also of conditions rarely occurring at the slope, large enough to ensure significance of the analyses carried out with ML techniques. In this respect, it is worth noting that the adopted clustering and Random Forest techniques allow easily handling big amounts of data without unaffordable computational burden.

### 2.2.1.  Definition of synthetic rainfall events

The Neyman-Scott rectangular pulse model (NSRP) has been used to obtain a 1000 years long synthetic hourly series of precipitation. The NSRP model reproduces the precipitation process as a set of rain clusters, composed by possibly overlapping rain cells embodied by rectangular pulses, each one with random origin. The storm duration is represented by the cell width and its height represents the associated rainfall intensity, so that when multiple cells overlap, the total intensity is the sum of the intensities of the overlapping cells (Rodriguez-Iturbe et al. 1987; Cowpertwait et al. 1996).

NSRP model calibration requires the identification of five parameters, using the method of moments (Peres and Cancelliere, 2014), based on available rainfall data for the investigated site. Specifically, the data from the rain gauge station of Cervinara, situated near the Loffredo village, belonging to the Civil Protection Agency of Campania Region available from January 2001 to December 2017 with a time resolution of 10 min, were used.

The aim of this study is the identification of variables expressing the slope conditions responsible of different responses to precipitation. In that sense, it is important to define the events within the rainfall time series to clearly distinguish antecedent conditions from the effects of the current rainfall event.

In other words, within the 1000 years long time series, a criterion should be identified to separate rainfall events, so that a new event begins only when the effects of the previous one disappeared. For this study, the events were defined as periods with at least 2mm of rainfall, preceded and followed by at least 24h with less than 2mm (i.e., smaller than the mean daily potential evapotranspiration estimated for the case study). Indeed, the separation period of 24 hours is commonly used for the definition of the empirical thresholds for early warning

systems against rainfall-induced landslides (e.g., Peres et al., 2018; Segoni et al.,
2018, Marino et al., 2020b).

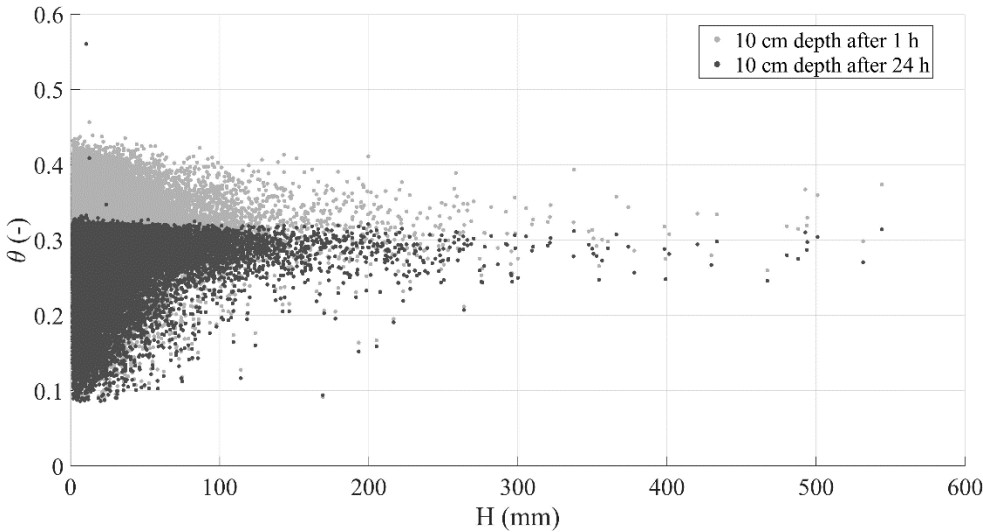


**Figure 4. Scatter plot of event rainfall depth and mean volumetric water content of**
**the top 10 cm soil depth 1 hour (grey dots) and 24 hours (black dots) after the end**
**of each rainfall event**
In fact, the mean volumetric water content ($\theta$) at 10 cm depth drops below soil
field capacity ($\theta \cong 0.35$) 24 hours after the end of each event (Figure 4) in all
the cases in which such value was overcome before the end of the event. This
shows that a dry interval of 24 hours after a rainfall event is long enough for
drainage processes to remove from the topsoil most of the water infiltrated from
the previous event. As topsoil moisture controls the infiltration capacity at ground
surface, after such interval the infiltration of new rainfall is only little affected by
the remnants of the previous rainfall event.
With the assumed separation criterion, a total of 53061 rainfall events within
1000 years are obtained, with durations ranging between 1 and 570 hours, and
total rainfall depth between 2 and 710 mm.

### 2.2.2. Slope hydrological model

As already pointed out in Section 2.1, the regular geometry of the slope, and the hydraulic characteristics of the soils, make the flow processes in the soil mantle mostly one-dimensional. Indeed, a simplified 1-D model had been previously developed and successfully validated according to the data collected during the hydrological monitoring activities (Greco et al., 2013; Greco et al., 2018), and was applied to investigate the hydrological response of the slope to synthetic hourly precipitation data. The unsaturated flow through the soil mantle is modelled with 1-D head-based Richards' equation (Richards, 1931), assuming for simplicity a single homogeneous soil layer, and it is coupled with a model of the saturated water accumulated in the perched aquifer. The adoption of a 1-D model is allowed thanks to the geometry of the considered mantle, as well as to the prevailing water potential gradients orthogonal to the ground surface when the soil is in unsaturated conditions.

The root water uptake has been accounted in the source term of the model, according to the expressions by Feddes et al. (1976), based on estimated potential evapotranspiration, with maximum root penetration depth equal to the soil mantle thickness and triangular root density shape.

Two boundary conditions are considered for the unsaturated soil mantle. At ground surface (i.e., the upper boundary condition), if the rainfall intensity is greater than the current infiltration capacity, the excess rainfall forms overland runoff. Otherwise, all rainfall intensity is set as infiltration. The bottom boundary condition links the soil mantle to a perched aquifer developing in the fractures and hydraulically connected to the unsaturated cover through the weathered soil layer (less conductive and capable of retaining much water), located at the contact between the cover and the bedrock. This soil layer penetrates the vertical conduits and fractures (Greco et al., 2013). In this context, the perched aquifer is modelled as a linear reservoir model, that receives water from the gravitational leakage of

the overlying unsaturated soil mantle and releases it as deep groundwater recharge and spring discharge (Greco et al., 2018). This conceptualization of the perched aquifer behaviour implies that the streamflow, supplied by the springs, is linearly related to the aquifer water level temporarily developing in the epikarst. Indeed, with this assumption, the model closely reproduces the trend of the stream water level observed in the field (Greco et al., 2018; Marino et al., 2020a). The pressure head at the soil–bedrock interface is assumed to follow the fluctuations of the water table of the underlying aquifer.

The hydraulic parameters of the homogeneous soil mantle have been obtained considering the information from previous laboratory tests (Damiano and Olivares, 2010) and field monitoring data analysis (Greco et al., 2013), considering the van Genuchten-Mualem model for the hydraulic characteristic curves (van Genuchten, 1980). Specifically, the parameters of the hydraulic characteristic curves were searched with a Genetic Algorithm, constrained within intervals ensuring the obtained curves to resemble available measurements of water retention and unsaturated hydraulic conductivity, obtained both in the field and in the laboratory (Greco et al., 2013). The parameters describing the hydraulic behaviour of the perched aquifer hosted in the upper part of the limestone bedrock have been derived from previous studies, which showed that the model satisfactorily reproduced the fluctuations of water potential and moisture, observed at various depths in the unsaturated soil cover, both during rainy and dry seasons (Greco et al., 2013; 2018). Model parameters are summarized in Table 1. The groundwater level of the perched aquifer is referred to the base of the epikarst, which is assumed 14 m below the soil-bedrock interface.

**Table 1. Hydraulic parameters of the coupled model of the unsaturated soil mantle and of the aquifer hosted in the epikarst** (Greco et al. 2021)**.**

| | | |
|---|---|---|
| Soil mantle | Soil mantle thickness (m) | 2 |
| | Saturated water content (-) | 0.75 |
| | Residual water content (-) | 0.01 |
| | Air entry value (m$^{-1}$) | 6 |
| | Shape parameter (-) | 1.3 |
| | Saturated hydraulic conductivity (m/s) | $3 \times 10^{-5}$ |
| Epikarst | Epikarst thickness (m) | 14 |
| | Effective porosity (-) | 0.005 |
| | Time constant of linear reservoir (days) | 871 days |

The equations have been numerically integrated with the finite difference technique, with a time step of 1 hour over a spatial grid with vertical spacing of 0.02 m.

The model assumes a homogeneous soil profile and a simplified slope geometry, and indeed it is not aimed at reproducing the details of flow processes through the unsaturated soil mantle. Consequently, the hydraulic properties of the homogeneous soil layer should be considered as effective properties, useful to reproduce the major features of the infiltration and drainage phenomena. The model is rather used to assess how large-scale (in time and space) hydrological processes, such as long-term cumulated rainfall and evapotranspiration and perched aquifer recharge, control the conditions that affect the response of the soil mantle to precipitation events. In this sense, the obtained results can be considered representative for large areas that share the major geomorphological features of the slopes of Partenio Massif.

### 2.2.3.  Synthetic hydrometeorological data

As it has been stated from previous sections, the dataset comes from the simulation of the hydrologic response of a slope to 1000 years long hourly rainfall time series, carried out with a physically based model, calibrated for the case

study. The output contains the time series of soil water content and suction at all depths throughout the soil mantle, of the water exchanged between the soil and the atmosphere, of the leakage through the soil-bedrock interface, and of the predicted water level of the underlying aquifer.

One hour before the onset of each rainfall event, the following variables have been extracted, as they would be measurable in the field and are representative of antecedent conditions: the aquifer water level ($h_a$), the mean volumetric water content in the uppermost 6 cm of soil mantle ($\theta_6$) and the mean volumetric water content in the uppermost 100 cm of soil mantle ($\theta_{100}$). To quantify the effects of rainfall on the slope response, the change of the water stored in the soil mantle at the end of each rainfall event ($\Delta S$) has been computed and compared with the total rainfall depth of the event (H).

Specifically, the inclusion of soil water content information has been chosen, as it can be obtained from available satellite-derived remote sensing products (Paulik et al., 2014; Pan et al., 2020) or from field sensor networks (Wicki et al., 2020). Regarding satellite products, in many cases not giving precise water content values, they satisfactorily reproduce temporal trends, which represent a valuable information for hazard assessment.

Besides, as the model introduces a linear relationship to estimate the outflow from the groundwater system, the monitored stream water level has been considered interchangeable with the simulated groundwater level, as the two variables are assumed directly linked in the model.

### 2.3. Data analysis techniques

The resulting dataset has been analyzed with Machine Learning techniques, aiming at capturing the complex interactions between the hydrological subsystems (i.e., soil mantle, fractured bedrock, surface water). Indeed, the analysis of the data is not only constrained to classical statistical analyses, such

as data frequency distributions, but also to data classification based on their geometrical distribution, and on quantifying the importance of the considered antecedent variables on the simulated response as well.

### 2.3.1. Variable importance assessment by Random Forest

Aim of this study is to find a set of measurable variables which, based only on field measurements, provide valuable information for predicting the response of the soil mantle to precipitation. In this respect, a suitable tool is represented by Random Forest (RF), a Machine Learning method that sets its basis on the theory of regression/classification trees, bagging data and capturing even the complex or non-linear interactions in-between the data of a set with relatively low bias (Breiman, 2001). This method is often used to forecast a desired variable based on predictor variables in terms of regression or classification set of randomly constructed trees. RF analysis of importance allows quantifying how informative the input variables are to make good predictions of the output, which should not be confused with the information provided by a variance-based Sensitivity Analysis (SA). In fact, this latter, always based on a mathematical model linking input variables to output, explains how the variability of the output is related to the variability of the inputs, regardless how the output of a model resembles available observations. As in this case the analysed data set is synthetic, i.e., it has been obtained through a mathematical model, the results of a variance-based SA will also be presented, allowing to compare the different kind of information provided by the two analyses.

In this case, a regression based Random Forest technique is applied to predict the soil storage response ($\Delta S$) at the end of each rainfall event of total depth H, using as predictors all possible triplets of variables described in the section 2.2.3 (H, $h_a$, $\theta_6$ and $\theta_{100}$). Specifically, four Random Forest models have been developed: RF1 with input features $\langle H, \theta_6, h_a \rangle$, RF2, with input features $\langle H, \theta_{100}, h_a \rangle$, RF3, with as input features $\langle H, \theta_6, \theta_{100} \rangle$ and RF4 with input features: $\langle H, \theta_6, \theta_{100} \rangle$. The

80% of the dataset was used to train the models and tuning the major
hyperparameters of random forest algorithm: the number of trees, the maximum
depth, the minimum sample leaf, and the maximum number of feature (more
details about the evaluation and optimization of the hyperparameters are provided
in Appendix B).
Then, the best predictor triplet of variables is selected according to the lowest
value of the Root Mean Squared Error (RMSE) calculated using the test data set
consisting of the 20% of the remaining data.
Furthermore, to understand how a single predictor variable affects the regression
model, the importance of input variables (features) in the Random Forest
regression model has been assessed through the mean decrease in impurity
(Breiman, 2001), which is a measure of the ability of the tree to split the dataset
in classes. Impurity is here computed as the mean decrease of RMSE, when a
particular variable is used for splitting nodes across all the trees in the RF.
Specifically, RMSE is employed to assess the quality of splits, and to determine
the importance of features in predicting output values.
### 2.3.2.  Data classification by clustering analysis
The exploratory analysis of spatial large datasets is often performed by means of
clustering techniques, aiming at identifying different classes in the data,
accounting on the distribution of the variables under study. There are two types
of clustering algorithms used for class identification purposes: algorithms based
on the density of points and algorithms based on the distance between points. The
algorithm used here is named k-means, and it is a distance-based procedure to
cluster data, based on the number of desired clusters and their centroids. The
algorithm assigns every element in the dataset to a cluster, iteratively minimizing
the variance of the Euclidean distance of the elements of each cluster from their
centroids. Consequently, the data labelling is done based on their geometrical
disposition in the dot cloud, depending on the target number of clusters to be
identified (Lloyd, 1982; Arthur and Vassilvitskii, 2007). When variables with
very different magnitudes are being related for clustering purposes, it is
convenient to normalize the data keeping the relative distances between
observations. Therefore, the clustering here is applied to the standardized data to
exploit the variance of each variable and keeping the geometrical disposition
between observations stable.
As the k-means algorithm does not automatically estimate the optimal number of
clusters to be identified within the dataset, the Silhouette metric has been used
here to evaluate the preferred number of clusters (Rousseeuw, 1987; de Amorim
and Hennig, 2015). In fact, this metric quantifies the quality of cluster
identification by scoring the difference between the overall average intra-cluster
distances and the average inter-cluster distances related to the maximum between
the latter two. In that way the metric would always be a value ranging from -1
and 1, where typically 1 means that clearly distinguished clusters have been
identified, 0 means that the identified clusters are indifferent, and -1 means that
data are mixed in the identified clusters.
## 3. Results and discussion
The analysis is carried out on both field monitored and synthetic datasets, to
quantify the information provided by the defined antecedent variables useful to
predict the seasonal changes of the slope response to precipitation. The analysis
of the physical behavior of the studied slopes is based on the results of model
simulations, as if they satisfactorily resemble what could be measured in the field.
Indeed, the uncertainty of model parameters may affect the identified cause-
effect relationships. However, during the calibration of model, field
measurements of the hydraulic behavior of the involved soil were considered
(Greco et al., 2013), thus the major features of the hydrological processes
occurring in the slope are considered reliably reproduced in the synthetic dataset.

## 3.1. Role of measurable variables on the response of the soil mantle

To select the most informative triplets of variables, for predicting the change in water storage ($\Delta S$) in the soil mantle, associated to rainfall events of total depth H, four Random Forest models are trained to predict the ratio $\Delta S/H$, based on the dataset consisting of all possible combinations of the synthetic variables: $\langle H, \theta_6, h_a \rangle$, $\langle H, \theta_{100}, h_a \rangle$, $\langle H, \theta_6, \theta_{100} \rangle$ and $\langle \theta_6, \theta_{100}, h_a \rangle$. In fact, the change in storage $\Delta S$ is obviously strongly dependent on the event rainfall depth H (i.e., the more it rains the more soil storage increases), thus concealing important hydrological processes going on the slope. Differently, the choice of the ratio $\Delta S/H$, a measure of the amount of rain that remains stored in the soil mantle, allows detaching the water drainage processes from the water accumulation processes. For each Random Forest model, the values of the Root Mean Square Error (RMSE) are calculated, and the importance of each predictor variable is evaluated according to the procedure described in Section 2.3.1. The computational effort implied in doing the calculations by a conventional workstation with a Core(TM) i7-10870H processor and 16 GB of SDRAM memory is less than 2 minutes for each model run. The obtained results are reported in Table 2.

**Table 2. RMSE and variable importance for H, $\theta_6$, $\theta_{100}$ and $h_a$ in the prediction of soil response described as $\Delta S/H$**

| Dataset | RMSE | Importance | | | |
|---|---|---|---|---|---|
| | | H | $\theta_6$ | $\theta_{100}$ | $h_a$ |
| $\langle H, \theta_6, h_a \rangle$ | 0.122 | 0.156 | 0.140 | - | 0.704 |
| $\langle H, \theta_{100}, h_a \rangle$ | 0.120 | 0.143 | - | 0.164 | 0.693 |
| $\langle H, \theta_6, \theta_{100} \rangle$ | 0.140 | 0.287 | 0.440 | 0.273 | - |
| $\langle \theta_6, \theta_{100}, h_a \rangle$ | 0.124 | - | 0.101 | 0.133 | 0.766 |

All the choices of triplets indicate that all the tested variables are informative to predict the normalized soil mantle response $\Delta S/H$ (Table 2), with the perched ground water level, $h_a$, resulting the most influent variable. The importance of $h_a$ on the response of the soil mantle suggests that, in some conditions, the change in soil storage is affected by the effectiveness of water exchange between the soil mantle and the underlying aquifer, as it will be discussed in the following sections. Moreover, in Table 2 the triplet showing the lowest RMSE values is formed by the total rainfall depth, the aquifer water level, and the mean volumetric water content in the uppermost 100 cm. According to the Random Forest model, they are the most informative for predicting the soil mantle response. Therefore, the triplet $\langle H, \theta_{100}, h_a \rangle$ is used for further analysis.

Considering the triplet of input variables $\langle H, \theta_{100}, h_a \rangle$, a variance-based Sensitivity Analysis has been also carried out, based on the methodology outlined by Sobol (2001), which is implemented in the Sensitivity Analysis Library in Python - SALib toolbox (Herman and Usher, 2017; Iwanaga et al., 2022). The sampling scheme proposed by Saltelli (2002) has been used to generate 65536 triplets, so to have a similar number of data as for the RF importance analysis. Table 3 reports the obtained sensitivity indices.

**Table 3. Sensitivity indices of the variance-based SA of the variability of $\Delta S/H$ resulting from variations of H, $\theta_{100}$ and $h_a$**

| Variable | $S_{tot}$ | $S_1$ (single parameter variations) | $S_2$ (mutual interactions) | |
|----------|-----------|-------------------------------------|------------------------------|--|
| $\theta_{100}$ | 0.532 | 0.471 | $(\theta_{100}, h_a)$ | 0.002 |
| $h_a$ | 0.058 | 0.058 | $(\theta_{100}, H)$ | 0.060 |
| $H$ | 0.469 | 0.412 | $(h_a, H)$ | 0.000 |

Interestingly, the indices show how the aquifer water level, $h_a$, which is the most
informative variable for output predictions according to the RF analysis, is
responsible only for a small part of the output variability, which instead is mostly
related to the variations of the other two input variables. As it will be discussed
in sections 3.2 and 3.3, $h_a$, not affecting the variability of $\Delta S/H$, is anyway an
extremely informative variable, as it allows separating the initial conditions in
two families: low levels and high levels, corresponding to quite different
responses of the soil mantle to precipitation. It also arises that output variability
mostly depends on the variations of single inputs (i.e., the indices $S_1$ explain most
of the total sensitivity, and the indices $S_2$, measuring the contribution to the total
output variance deriving from mutual interactions between couples of inputs are
all small).
### 3.2.    Soil and underground antecedent conditions
The field monitoring activities allow to get a complete dataset that traces the
rainfall values coupled with the soil mean volumetric water content in the
uppermost meter of the soil profile ($\theta_{100}$) and the water depth of the Castello
stream ($h_s$), both measured hourly for three years. The field monitored data,
composed by 57 rainfall events, include the water level of the Castello stream
rather than the direct measurement of the aquifer water level ($h_a$). Nevertheless,
a direct relationship links the water level in the aquifer and the water level in the
stream, as assumed for the mathematical modelling. This dataset has been
enriched synthetically, as it has been described in section 2.2.

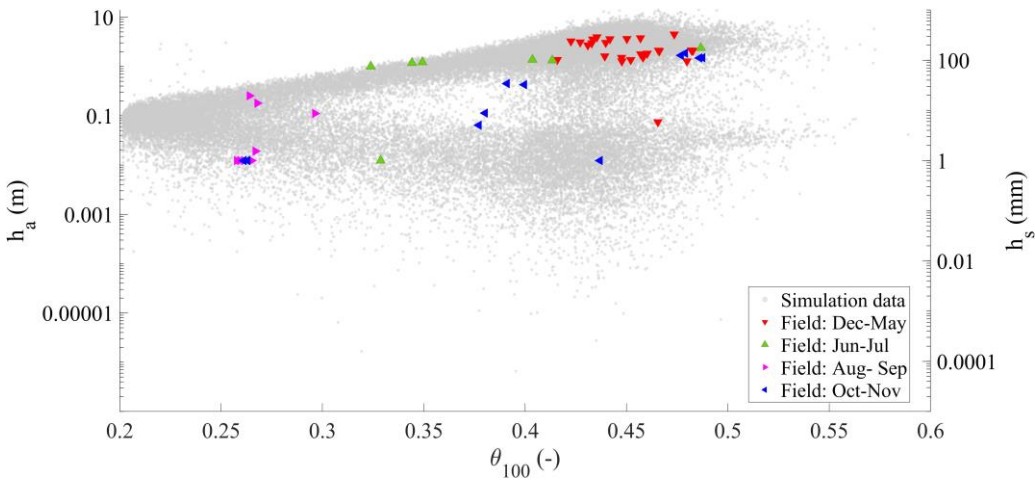


**Figure 5. Field monitored mean volumetric water content in the upper meter of the**
**soil profile ($\theta_{100}$) and water depth in the Castello stream ($h_s$), compared with**
**synthetic data of $\theta_{100}$ and aquifer water level ($h_a$) (on the vertical axis, plotted in**
**logarithmic scale to help visualizing of small water levels and thus not allowing to**
**represent zeroes, the values of $h_s$ smaller than the sensitivity of the water level**
**sensor have been plotted as 1 mm; also the smallest simulated values of $h_a$ should**
**be considered equivalent to zero, owing to the limits of any measurement device,**
**which could be used for operational field monitoring).**
Therefore, to analyze the effects of the underground conditions on the slope
response, Figure 5 shows the simulated data (circular dots in the background) and
the field monitored data (triangular colored dots). Logarithmic axes are used to
distinguish the very low aquifer water level from the high values.
Four major seasonally recurrent conditions could be identified for the water in
the subsurface system from field monitored data: first, a condition usually
occurring between December and May is characterized by the highest water
content in the soil and the highest measured water level in the stream. Second,
the period from June to July is characterized by intermediate water content
values, with still high level in the stream. Third, the period from August to
September is characterized by the lowest values of water content in the soil, but
also the lowest water depth $h_s$ measured in the stream (few centimeters, in some
cases nearly zero). Finally, the period from October to November is characterized
by a wide range of values in soil water content and a relatively low range of
stream water depth.

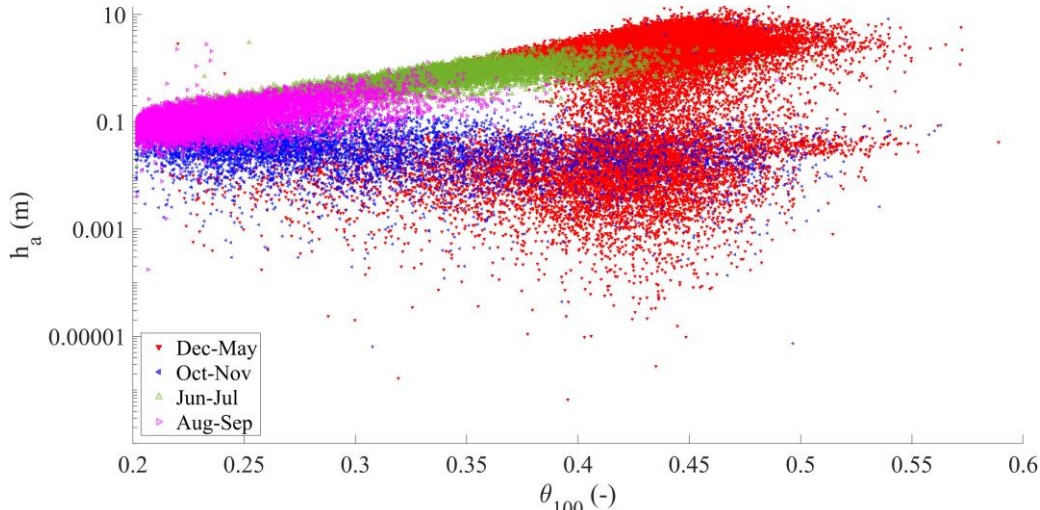


**Figure 6. Seasonal behavior of the aquifer water level ($h_a$) and the mean volumetric**
**water content of the upper meter of the soil profile ($\theta_{100}$) for the synthetic dataset**
**(the vertical axis is plotted in logarithmic scale to help visualizing small water**
**levels).**
The underground antecedent conditions are naturally linked to a seasonal
behavior dominated by the hydrological conditions which can be traced in time
as it can be seen from the synthetic data (Figure 6). The months from December
to April follow a winter and spring behavior, characterized by wet soil conditions
and aquifer water levels ranging from low to high. From June to July, a late spring
behavior is visible, characterized by relatively dry soil (i.e., most of the data
falling below soil field capacity), in combination with relatively high
groundwater levels (indicating a still active slope drainage). In August and
September, a summer like behavior is shown, with the driest soil water content
and generally low aquifer water level. Finally, in October and November, the end
of the dry season is shown: a wide range of soil wetness coupled with a still low
aquifer water level.
For both the field monitored and synthetically obtained datasets, the observed
conditions are the result of the time lag between the beginning of the rainy season
and the slope response. The recurrent seasonal behavior observed for the
synthetic dataset, although delayed or anticipated owing to the year-by-year
variability of rainfall, is close to that observed in the field.

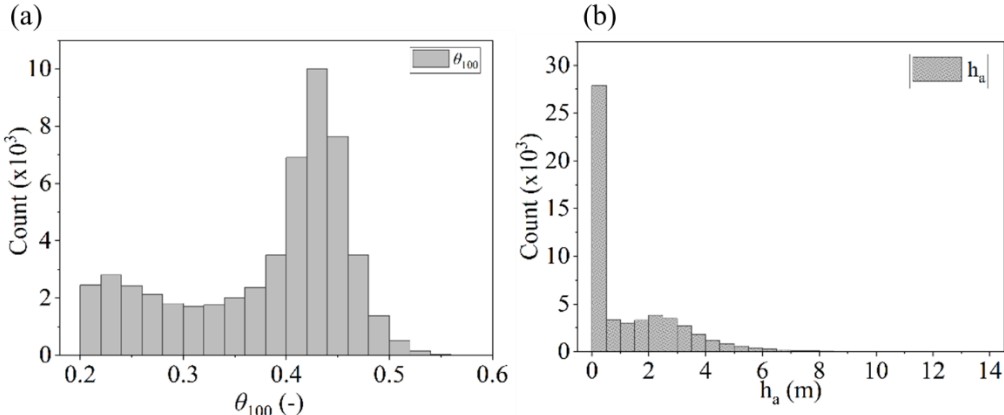


**Figure 7. Histograms for data distributions of (a) $\theta_{100}$ and (b) $h_a$ for the synthetic**
**dataset**
The overall situation for the synthetic dataset of antecedent conditions (i.e.,
duplets $\langle \theta_{100}, h_a \rangle$) can be described by the distribution of each individual
variable, which can be seen in the histograms shown in Figure 7. It is interesting
to note that, for both $\theta$ and $h_a$, a bimodal behaviour is observed, corresponding
to dry and wet field conditions.

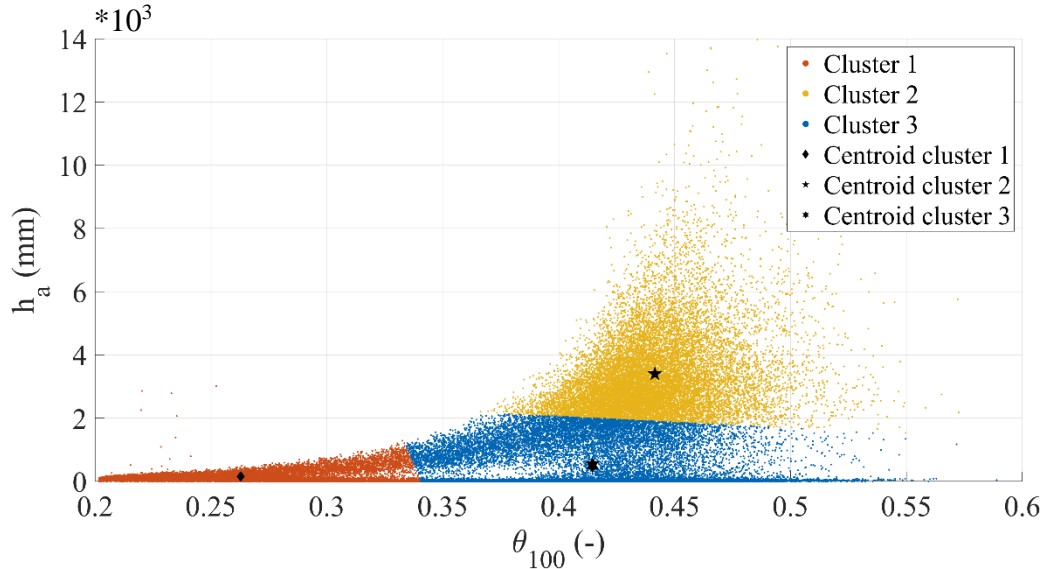


**Figure 8. Identified clusters for the duplets $\langle \theta_{100}, h_a \rangle$ representing underground antecedent conditions of the synthetic dataset. For each cluster, the centroids are shown.**

The k-means clustering technique has been used to investigate the geometrical distribution of the duplets $\langle \theta_{100}, h_a \rangle$, with number of clusters ranging from 2 to 7. According to the Silhouette metric, the optimal number of clusters is 3, with a metric value of 0.7, allocating the 28%, 30% and 42% of the data in clusters 1, 2 and 3 respectively. Figure 8 shows the 3 clusters obtained within the synthetic dataset. Centroid positions are also displayed, showing the zones of the clouds where most of the dots are gathered. This representation of the data use both vertical and horizontal axes in linear scale to let visualize distance magnitudes between the different clusters, but it corresponds to the same dataset shown in Figure 6.

The distribution of the data after clustering is also analyzed for each cluster and the histograms are shown in Figure 9. It looks clear that the clusters capture different couplings of dry and wet underground antecedent conditions.

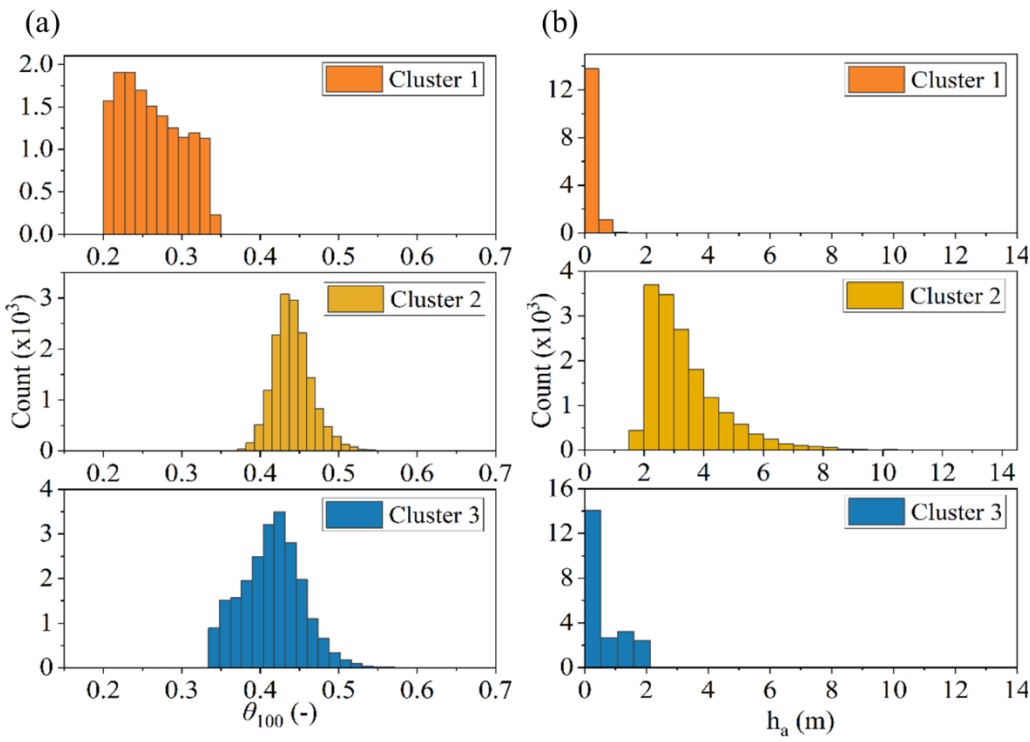

**Figure 9. Histograms for data distributions of (a) $\theta_{100}$ and (b) $h_a$, according to each identified cluster in the duplets $\langle \theta_{100}, h_a \rangle$**

In fact, cluster 1 captures dry conditions, with a volumetric water content below the field capacity $\theta_{fc}$ (it was estimated as 0.35 with the empirical relationship proposed by Twarakavi et al. (2009) according to the van Genuchten model parameters) and low values of $h_a$. Differently, clusters 2 and 3 capture scenarios related to relatively wet soil mantle conditions (i.e., $\theta_{100} > \theta_{fc}$), coupled to low $h_a$ in cluster 3, gathering scenarios normally observed in late autumn, and to the highest $h_a$ conditions for cluster 2, comprising conditions normally occurring in late winter and spring.

The two chosen variables, $\theta_{100}$ and $h_a$, allow identifying three different antecedent slope conditions one hour before the onset of any rainfall event. Hence, it is worthy to investigate how these different antecedent conditions may be related to different slope responses to precipitation.

     **3.3.**     **Effects of soil and underground antecedent conditions on the**
                **slope response to rainfall**

The analysis of the data has been focused on identifying clusters within the
triplets $\langle \theta_{100}, h_a, \Delta S/H \rangle$, aiming to evaluate the slope response as the amount of
rainwater being stored/drained in the soil mantle. The results are being plotted in
the space composed by the variables that can be monitored in the field:
$(\theta_{100}, h_a, H)$.
As it is not always expected to experience increased soil storage during rainfall
events, the identification of draining slope conditions is an important aspect.

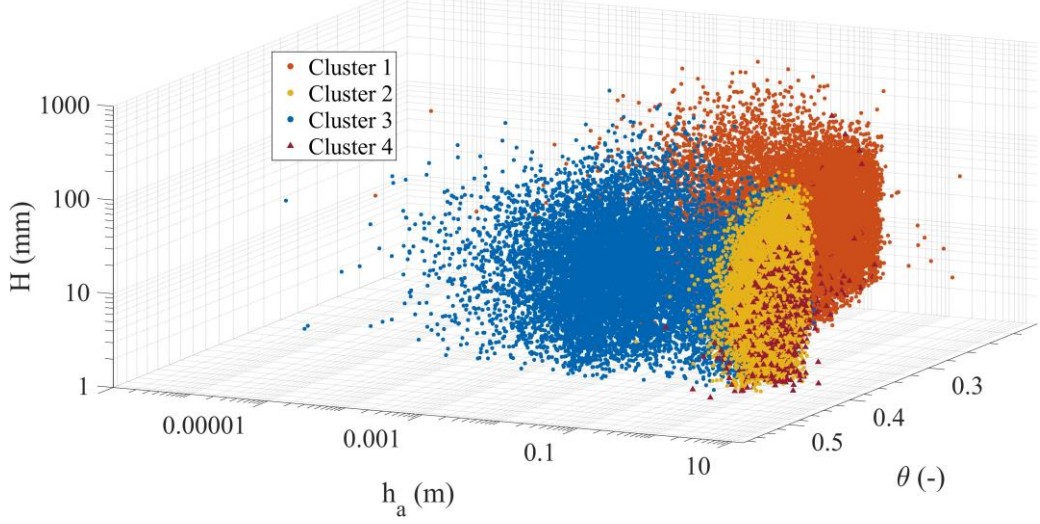


**Figure 10. Clustering results of the synthetic data triplets $\langle \boldsymbol{\theta_{100}}, \mathbf{h_a}, \boldsymbol{\Delta S/H} \rangle$**
**represented in the space $(\boldsymbol{\theta_{100}}, \mathbf{h_a}, \mathbf{H})$**

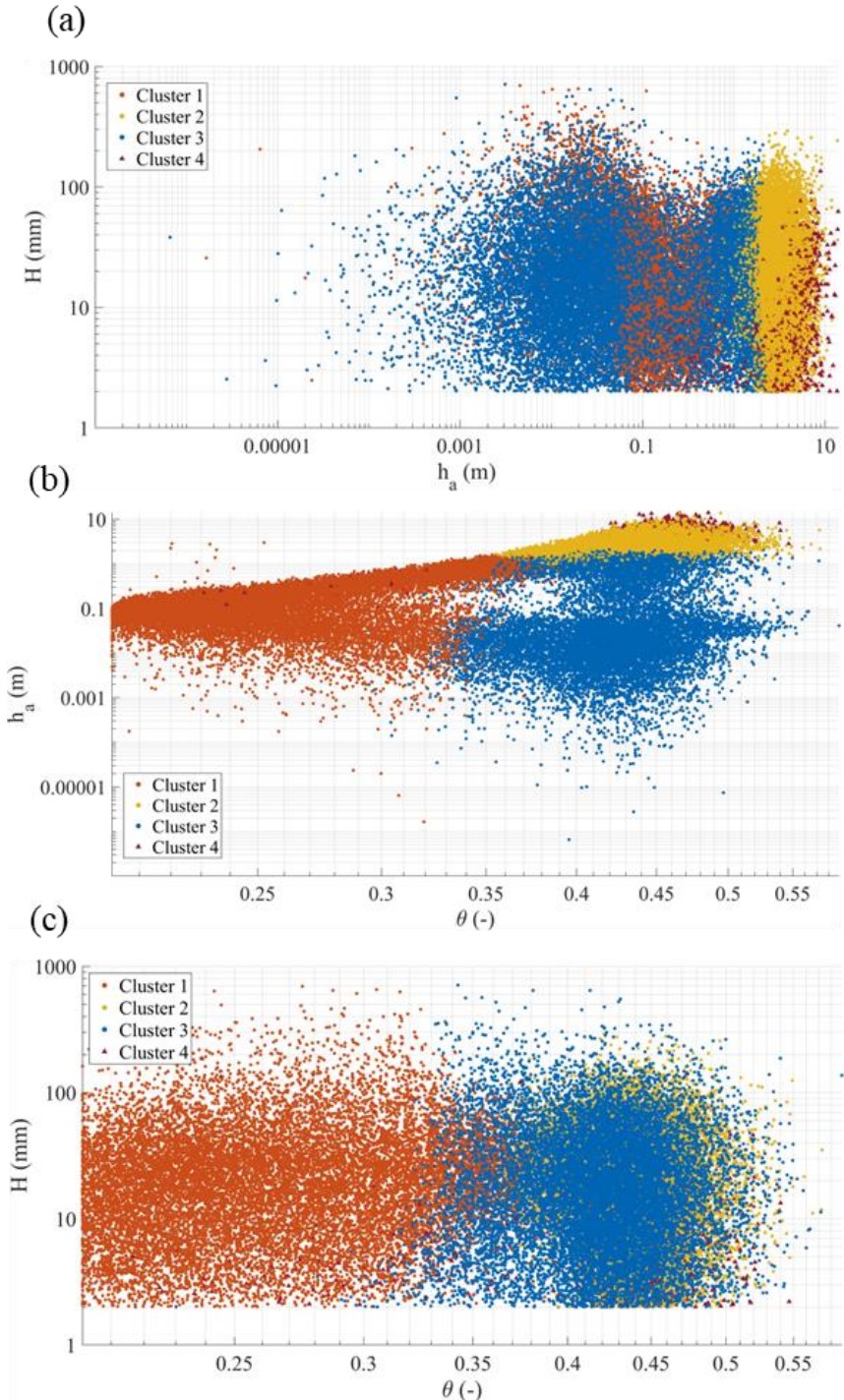


Figure 11. Clustering results of the triplets $\langle \theta_{100}, h_a, \Delta S/H \rangle$ in (a) $(\theta_{100}, h_a)$ plane;
(b) $(\theta_{100}, H)$ plane; (c) $(H, h_a)$ plane

Figure 10 and Figure 11 show the data clusters for the triplets $\langle \theta_{100}, h_a, \Delta S/H \rangle$,
for any identified rainfall event, represented in the $(\theta_{100}, h_a, H)$ space in a
logarithmic axis representation. The Silhouette metric in this case suggests 4 as
an optimal number of clusters with a metric value of 0.61. It is remarkable that
three of the clusters are close to those already identified from the antecedent
(seasonally recurrent) underground conditions (section 3.2).

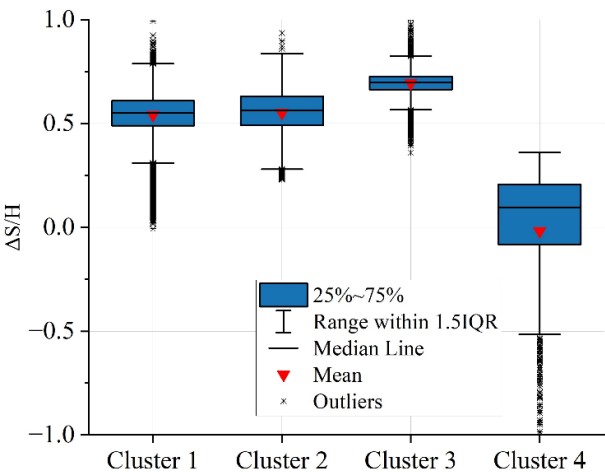


**Figure 12. Distribution of the slope response ΔS/H for the data in each cluster**
Specifically, cluster 1, 2 and 3 correspond to different slope processes according
to ΔS/H (Figure 12). Even if cluster 1 and cluster 2 show similar responses, with
slightly smaller ΔS/H for cluster 1, the controlling processes are indeed different;
the conditions of cluster 1 are typically occurring in dry seasons with long dry
periods between short rainfall events, leading to dry antecedent conditions, so
that accumulation of water in the soil mantle (increase in water storage) is
expected at each event. The data in cluster 2 are typically related to wet seasons,
especially in late winter and spring, where rainfall events are more frequent,
leading to antecedent wet soil $(\theta_{100} \geq \theta_{fc})$ and antecedent high ground water
level. However, these conditions do not seem to correspond to effective slope
drainage, so that the slope response in cluster 2 results comparable to that
observed in cluster 1 in terms of ΔS/H. Instead, the conditions gathered in cluster
3 differ from those in cluster 2 for the lower aquifer water level $h_a$, and the
highest $\Delta S/H$ indicates the lowest slope drainage.
The additional cluster 4 identified here highlights a particular slope response, as
it catches all the conditions where nearly zero and negative $\Delta S$ take place,
meaning an effective slope drainage during rainfall events. It is interesting to note
that, even for relatively high rainfall events (above 100 mm), this slope response
occurs when soil moisture is above the field capacity and when this condition is
coupled with very high groundwater level, probably due to the high permeability
all along the soil mantle and to the hydraulic connection with the underlying
aquifer.
**4.     Conclusions**
This study aims at identifying and analysing the major hydrological controls of
the slope response to precipitation and, in that way, defining suitable variables to
be monitored in the field to predict such response. The studied case refers to the
hydrological processes in a slope system consisting of a pyroclastic soil mantle
overlaying a fractured karstic bedrock, where a perched aquifer develops during
the rainy season. A synthetic time series of slope response to precipitation has
been built, thanks to a physically based model, previously calibrated with field
monitoring data, coupled with a stochastic rainfall generator. Synthetic and
experimental data show substantial agreement. In fact, the soil water content
values measured in the field are close to those of the synthetic dataset.
Furthermore, the simulated epikarst water level shows similar seasonal behaviour
as the stream level records, indeed directly related with the discharge from the
epikarst aquifer. The synthetic dataset has been explored with Random Forest
and k-means clustering, to evaluate the slope response characterized as the
change in water stored in the soil mantle ($\Delta S$) during precipitation events with
rainfall depth H, starting from different underground antecedent conditions.
These were quantified through the mean volumetric water content in the
uppermost meter of soil mantle ($\theta_{100}$) and the aquifer water level ($h_a$), one hour
before the onset of rainfall.
The ratio $\Delta S/H$, which allows identifying soil mantle response regardless the
amount of event precipitation, is sensitive to both $h_a$ and $\theta_{100}$, with the
groundwater level being the most influential antecedent variable. The
underground antecedent conditions, characterized by $\theta_{100}$ and $h_a$ and linked to
the seasonal meteorological forcing, allow identifying different responses,
related to the seasonally active hydrological processes.
High perched groundwater level, typical of winter and spring, indicates active
drainage from the soil mantle, which compensates rainwater infiltration, so that
the soil storage remains stable, or even reduces, even after large rainfall events.
Differently, low perched groundwater level corresponds to impeded drainage.
When it occurs with initially dry soil mantle (typically in summer and early
autumn), it tends to retain all the infiltrated rainwater as increased soil storage.
When the soil mantle is already wet (i.e., above the field capacity) at the onset of
rainfall events, as it usually happens in late autumn and early winter, the increase
of soil storage is smaller, as the soil approaches saturation.
The presented results suggest that monitoring antecedent conditions, by
measuring suitable variables to identify the major hydrological processes
occurring in the slope in response to precipitation, can be useful to understand
such processes and to develop effective predictive models of slope response.
Therefore, the proposed methodology can be replicated also in other contexts and
be useful for several hydrologic applications: from the water supply towards
natural streams due to infiltrated water, to the hydric stress estimation in crops
(e.g., the centenary chestnut forests of the case study) especially in very dry
seasons, but also for the design of effective monitoring networks exploiting
geohydrological information for geohazard prevention (and early warning).

**Appendix A: Calibration of the Stochastic Rainfall Generator**

The Neyman-Scott Rectangular Pulse (NSRP) model (Neyman and Scott, 1958; Rodriguez-Iturbe et al., 1987; Cowpertwait et al., 1996) is here used as stochastic rainfall generator. The NSRP describes the process of point rainfall as a superposition of randomly arriving rain clusters, each containing several rain cells with constant intensity. The hyetograph within a cluster is obtained by summing the intensity of the various cells belonging to the cluster. It has been calibrated based on 17 years of experimental data (2000-2016) of rainfall depth at 10 min time resolution, recorded by the rain gauge managed by the Civil Protection in Cervinara (Southern Italy). The calibration has been carried out by minimizing, for rainfall aggregated at various durations, the difference between the following quantities, estimated by the model and calculated from the experimental data: mean, variance, lag 1 autocorrelation, probability of dry interval, probability of transition from dry-to-dry interval and probability of transition from wet-to-wet interval. The calibration procedure, based on the one proposed by Coptwertwait et al. (1996), is described in detail in Peres and Cancelliere (2014). To account for the seasonality of rainfall, these quantities have been calculated month by month in the experimental record (Figure A1), suggesting that the calibration of the NRSP model should be carried out separately for seven homogeneous periods (September, October, November, December-March, April, May-June, July-August).

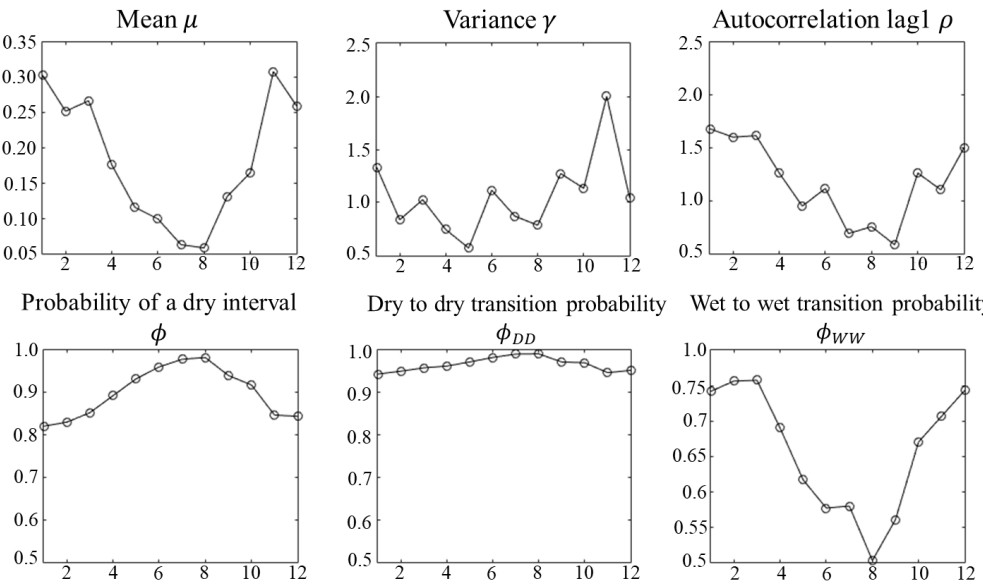


**Figure A1. Monthly plot of hourly rainfall characteristics calculated based on the experimental data of the rain gauge of Cervinara.**

Table A1 gives the obtained parameters of the NSRP stochastic model, where λ
represents the parameter of a Poisson process describing the arrival of clusters; ν
is the mean number of cells in a cluster, also described by a Poisson process; β is
the parameter of an exponential probability distribution describing the arrival
times of each cell in a cluster, expressed as the number of time intervals of 10
minutes starting from the beginning of a cluster; η is the parameter of an
exponential probability distribution describing the duration of rain cells; ξ is the
parameter of a Weibull probability distribution describing the rain intensity of
cells, with cumulative probability function $F(x, \xi, b) = 1 - \exp(-\xi x^b)$, in which
x is cell rain intensity and the parameter $b = 0.8$ has been set a priori
(Cowpertwait et al., 1996).



**Table A1. Parameters of the NSRP model.**

| Param. | Sept | Oct | Nov | Dec-Mar | Apr | May-Jun | Lug.Aug |
|---|---|---|---|---|---|---|---|
| $\lambda$ (h$^{-1}$) | 0.015 | 0.00524 | 0.00257 | 0.0238 | 0.00809 | 0.00386 | 0.00900 |
| $\nu$ (−) | 2.68 | 36.4 | 57.1 | 2.60 | 38.7 | 21.6 | 1.40 |
| $\beta$ (h$^{-1}$) | 0.265 | 0.156 | 0.0167 | 0.813 | 0.123 | 0.116 | 24.5 |
| $\eta$ (h$^{-1}$) | 1.41 | 57.3 | 1.43 | 0.280 | 15.5 | 8.59 | 1.23 |
| $\xi$ (h$^b$ mm$^{-b}$) | 0.330 | 0.047 | 0.450 | 0.967 | 0.186 | 0.158 | 0.268 |

The adherence of the rainfall generated with the stochastic model to the experimental rainfall data has been tested by evaluating rainfall characteristics different from those used for the calibration. For instance, Figure A2 shows the comparison of the rainfall depth, cumulated over one year, for the experimental data (17 years) and for 1000 years of synthetic data generated with the calibrated NSRP model.

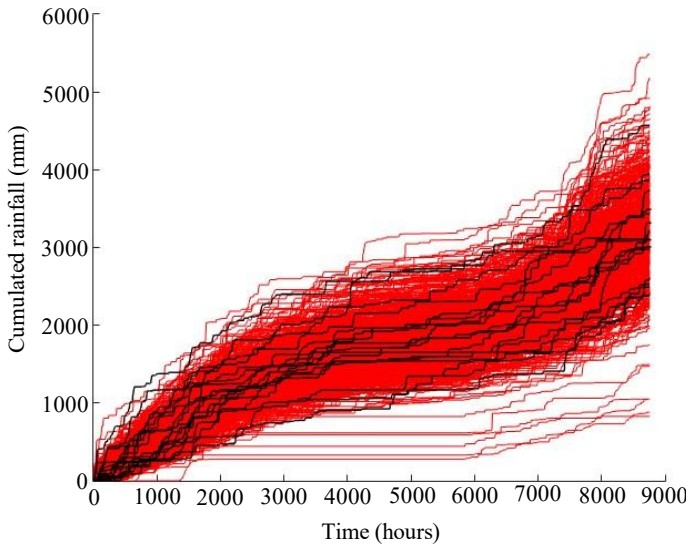

**Figure A2. Comparison of observed (black) and simulated (red) cumulated rainfall plots in a year.**

In Figure A3, the boxplot of the maximum hourly rainfall in one year, observed
in the experimental dataset of 17 years, is compared with the same boxplot
referred to 20 series of 17 years randomly extracted from the generated 1000
years synthetic rainfall series. Several of the synthetic 17 years intervals show a
distribution of the maximum hourly rainfall close to the observed one.

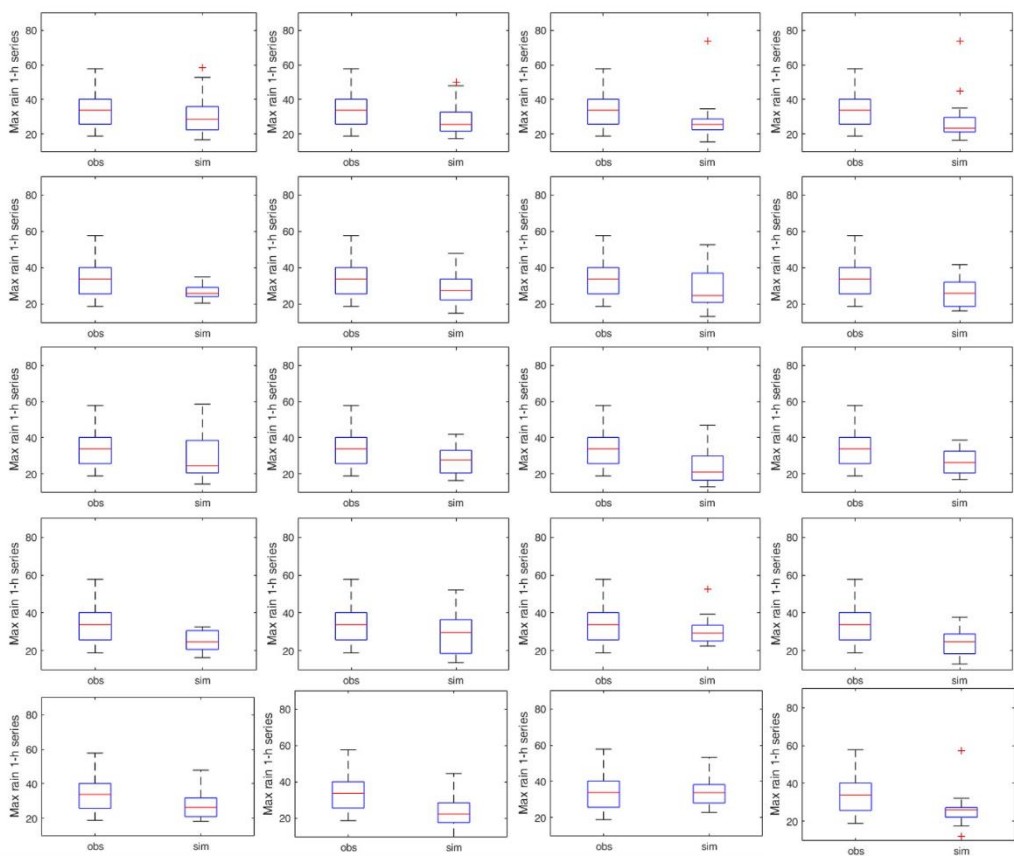


**Figure A3. Comparison of observed and simulated distributions (boxplots) of the**
**maximum hourly precipitation in a year, for series of the same length. Each panel**
**shows the distribution for the 17 observed years (boxplot is always the same), and**
**17 randomly picked simulated years.**

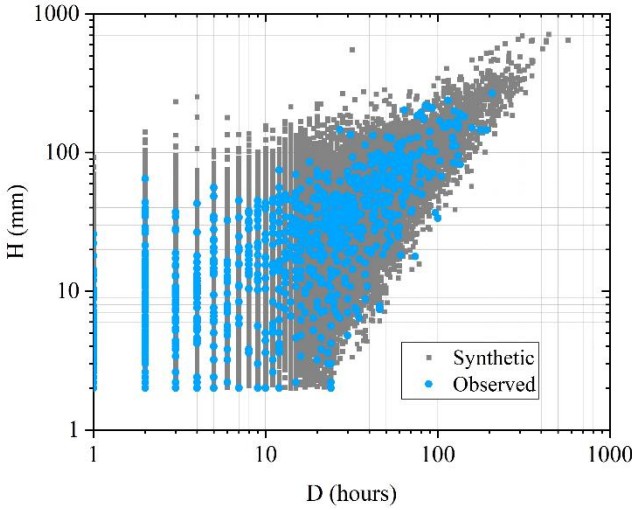


**Figure A4. Scatterplot of total rainfall event depth (H) vs. rainfall event duration (D). The events have been sorted within the rainfall datasets by considering a separation "dry" interval of 24 hours with less than 2 mm rainfall. The blue dots represent events extracted from the 17 years experimental rainfall dataset, while the grey dots represent events extracted from the 1000 years synthetic rainfall dataset.**

Regarding the required comparison between synthetic and observed wet and dry intervals, figure A4 shows the scatterplot of duration and total rain depth of the events, sorted with a separation "dry" interval of 24 hours with less than 2 mm rainfall from the observed dataset (blue dots) and the synthetic dataset (grey dots). The plots show how the synthetic data contain the observed ones, and that the shape of the dot clouds looks quite similar.

Figure A5 shows the frequency distributions of the durations of dry intervals belonging to the 17 years rainfall dataset, and the same distribution for the dry intervals extracted from the 1000 years synthetic dataset: the two distributions look nearly identical.

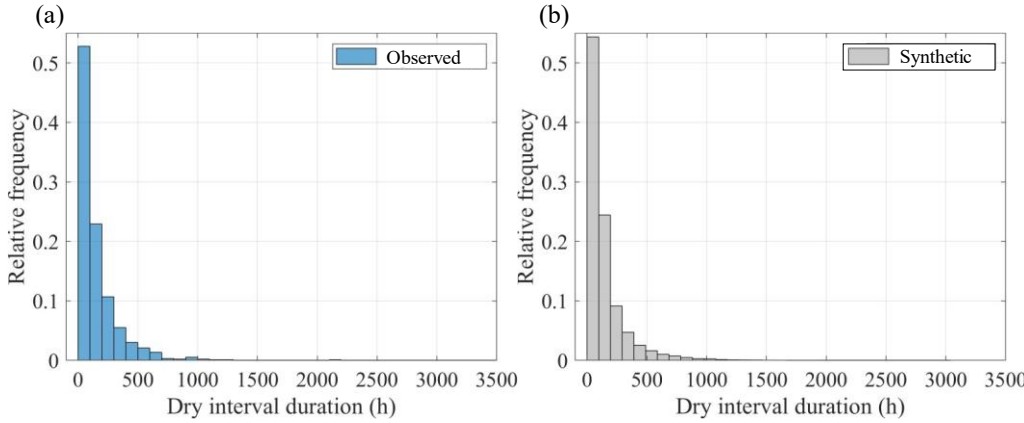


**Figure A5. Frequency distributions of dry interval durations for events extracted**
**from the 17 years experimental rainfall dataset (a) and events extracted from the**
**1000 years synthetic rainfall dataset (b). The events have been sorted within the**
**rainfall datasets by considering a separation "dry" interval of 24 hours with less**
**than 2 mm rainfall.**

**Appendix B: Tuning Random Forest hyperparameters**

The Random Forest (RF) algorithm (Breiman, 2001) has been very successful as a general-purpose classification and regression method. Starting from Bagging or Bootstrap Aggregation (Efron and Tibshirani,1993), RF builds several random de-correlated decision trees and then averages their predictions.

The regression RF algorithm can be summarized as follows: 1) by means of bootstrap, a sample is extracted from the training data; 2) based on the bootstrapped data, a tree $T$ of the random-forest is grown by repeating the following operations until a leaf node (a node without split) is reached: a) for each node, $m$ variables are randomly selected from the $p$ input variables or features (with $1 \leq m \leq p$); b) among the $m$ variables, the best variable and splitting point are selected according to a minimum criterium; c) the node is split into two daughter nodes. To build the RF with $B$ trees, steps 1 and 2 are repeated $B$ times. Then, the prediction, $Y_{pred}$, for a new observation, $X$, is the average of the final values, $T_b(X)$, i.e., the values of the predicted variable corresponding to the leaves of each tree:

$$Y_{pred} = \frac{1}{B}\sum_{b=1}^{B} T_b(X) \tag{B.1}$$

The main advantage of RF is the simplicity with which a forest can be trained, and the parameters of the algorithms optimized. In this paper, the scikit-learn framework (Pedregosa et al, 2011) is used to run the RF algorithm.

The main hyperparameters of a RF are: 1) n_estimators: the number of trees of the forest; 2) max_depth: the maximum depth of each decision tree in the forest; 3) min_samples_leaf: the minimum number of samples required to be at a leaf node; max_features: the number of features, or input variables, to consider when looking for the best split.

The procedure applied in this study to estimate and optimize the hyperparameters
of the RF algorithm consists of the following steps:
-    Step 1: the dataset is divided into a training set and a test set, respectively

867         containing 80% and 20% of the data, randomly chosen.

-    Step 2: the K-fold cross-validation technique (Stone, 1974), with K=10,

869         is applied to empirically determine a set of values for the

870         hyperparameters, using only the training dataset.

-   Step 3: for each fold, a RF is trained on the other k-1 folds of the data and
tested on the first fold. This process is repeated k=10 times, so to use each of
the k folds exactly once as the validation set. A performance metric is then
calculated for each fold, to estimate how well the RF will perform on new
data. In this work the Root Mean Square Error (RMSE) is used as the
performance metric.
-   Step 4: the RF is trained by changing one hyperparameters at once and using
the default values for the other three (default values of hyperparameters as
reported in Pedegrosa et al (2011) are: n_estimators=100; max_depth=*none,*
i.e., the tree is expanded until all leaves contain less samples than
min_samples_split; min_samples_leaf=1; max_features=1).
-   Step 5: from the results of the previous step, the ranges of hyperparameters,
given in table B1, are defined. These values represent the grid in which the
optimal hyperparameters are searched. In other words, using the K-fold
technique (step 2), RF model is fitted K times, and then the optimal set of
values is the one minimizing the RMSE.
-   Step 6 (validation of the model), once the optimal values of the
hyperparameters are determined, the performance of RF model is evaluated,
for the test dataset as defined in Step 1, using the RMSE.
In this study, the described methodology is used to evaluate the hyperparameters
for the following RF models: RF1, trained using the input features $\langle H, \theta_6, h_a \rangle$;
RF2, trained using $\langle H, \theta_{100}, h_a \rangle$; RF3, trained using $\langle H, \theta_6, \theta_{100} \rangle$; RF4, trained
using $\langle H, \theta_6, \theta_{100} \rangle$. All models are trained to predict the normalized change of
water storage in the soil mantle, $\Delta S/H$. Figures B1, B2, B3 and B4 show the
results of step 4. Specifically, they depict the trends of the RMSE versus the
hyperparameters for RF1, RF2, RF3 and RF4, respectively.

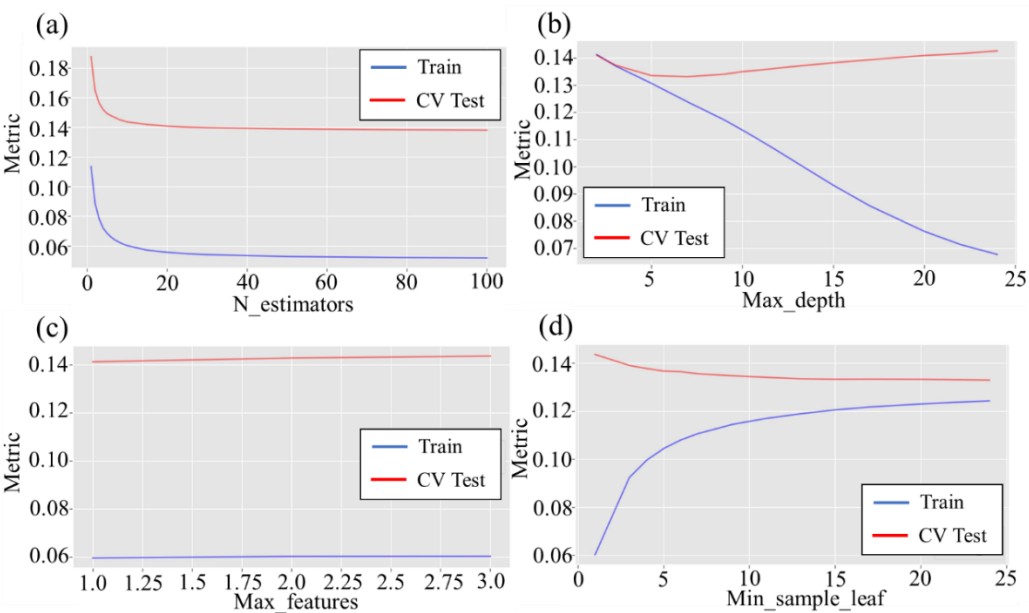


**Figure B1. Performance of random forest model RF1 on the test and Cross**
**Validation (CV) sets according to the test metric by changing the hyperparameters:**
**(a) N_estimators (b) Max_depth (c) Max_features (d) Min_samples_leaf**

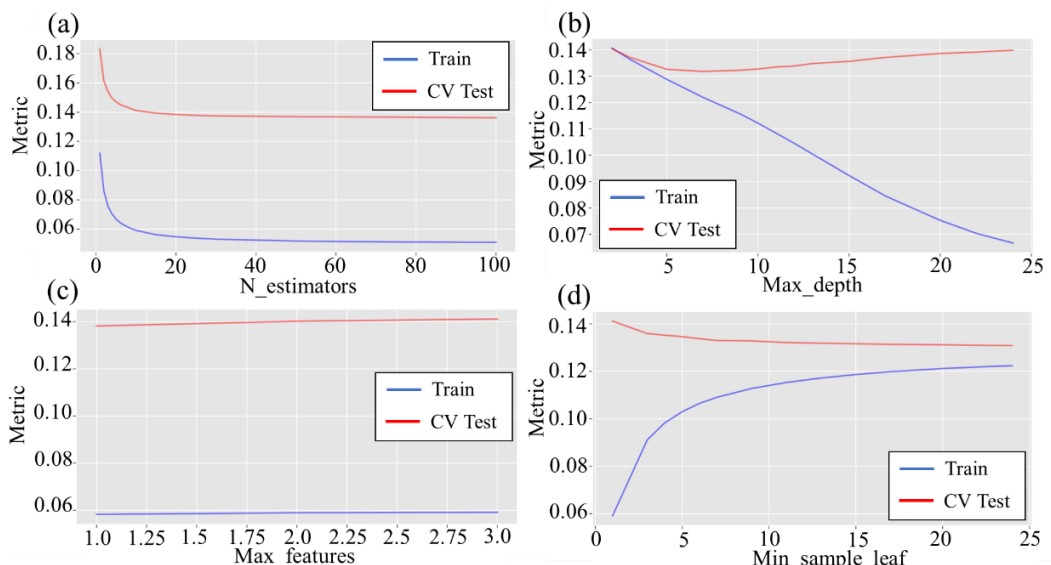


**Figure B2. Performance of random forest model RF2 on the test and Cross Validation (CV) sets according to the test metric by changing the hyperparameters: (a) N_estimators (b) Max_depth (c) Max_features (d) Min_samples_leaf**

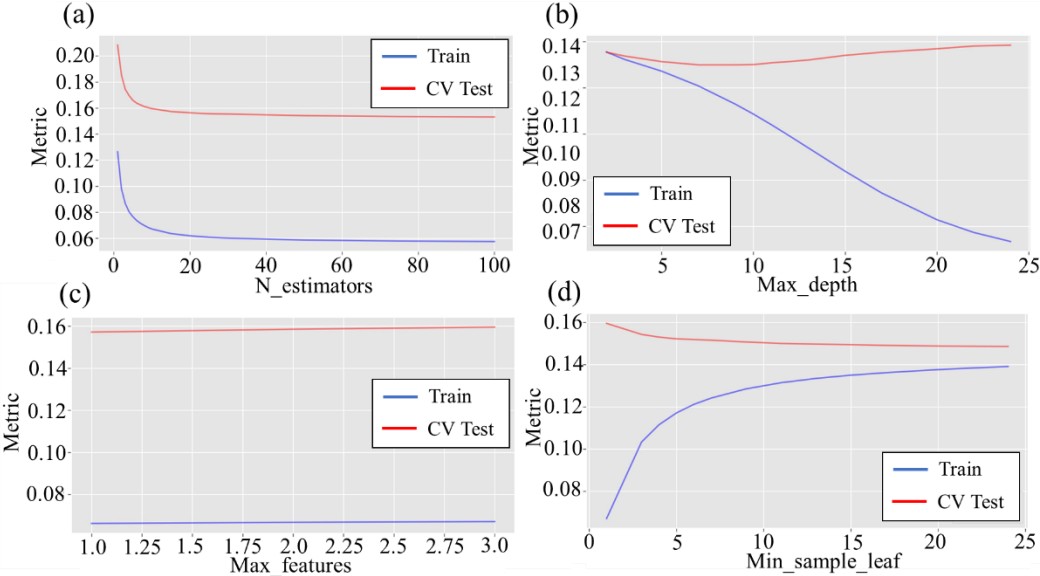


**Figure B3. Performance of random forest model RF3 on the test and Cross Validation (CV) sets according to the test metric by changing the hyperparameters: (a) N_estimators (b) Max_depth (c) Max_features (d) Min_samples_leaf**

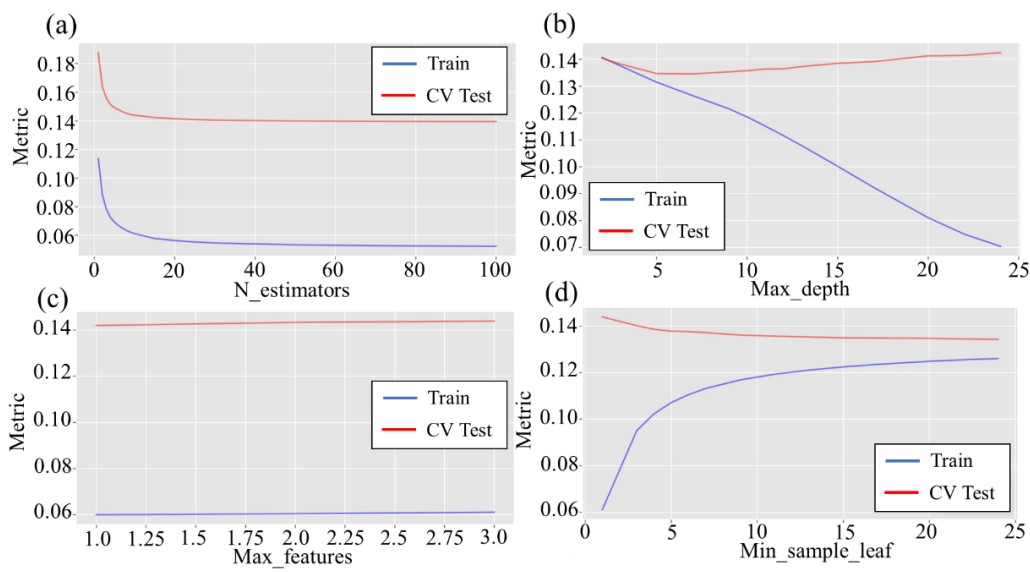


**Figure B4. Performance of random forest model RF4 on the test and Cross Validation (CV) sets according to the test metric by changing the hyperparameters: (a) N_estimators (b) Max_depth (c) Max_features (d) Min_samples_leaf**

The analysis of the previous figures provides the search gird of hyperparameters given in Table B1. After fitting each model K times (step 5), the optimal sets of hyperparameters are reported in Table B2 for each RF model. Then, the performance of models RF1, RF2, RF3, and RF4 are evaluated on the test dataset using RMSE metric. The obtained results are summarized in Table B3.

The above-described analysis has been used to identify the most informative triplet of variables, which has been chosen as the one corresponding to the best performing among the optimal RF models, namely RF2.

**Table B1. Hyperparameters range of variation**

| Hyperparameter | Range of variation |
| --- | --- |
| n_estimators | 5,10,20,25,30 |
| max_features | 1,2,3 |
| min_samples_leaf | 15,20,25 |
| max_depth | 3,4, 5, 6,7 |

**Table B2. Optimal values of Hyperparameters**

| Hyperparameter | Optimal values | | | |
|---|---|---|---|---|
| | **RF1** | **RF2** | **RF3** | **RF4** |
| n_estimators | 30 | 30 | 25 | 30 |
| max_features | 2 | 2 | 3 | 2 |
| min_samples_leaf | 20 | 20 | 9 | 20 |
| max_depth | 7 | 7 | 7 | 7 |


**Table B3. RMSE of studied models computed for the test dataset**

| Model | RMSE |
|---|---|
| RF1 $\langle H, \theta_6, h_a \rangle$ | 0.122 |
| RF2 $\langle H, \theta_{100}, h_a \rangle$ | 0.120 |
| RF3 $\langle H, \theta_6, \theta_{100} \rangle$ | 0.140 |
| RF4 $\langle \theta_6, \theta_{100}, h_a \rangle$ | 0.124 |




## Author contributions

RG and DR formulated the research aim; PM provided the field measurements; PM and GS supplied the model simulations; DR and GS curated and analyzed the data; RG oversighted the research activities; DR worked on the preparation and the data visualization; DR, PM and GS wrote the draft manuscript; RG wrote the final version of the manuscript.

## Acknowledgements

This research is part of the Ph.D. project entitled "Hydrological controls and geotechnical features affecting the triggering of shallow landslides in pyroclastic soil deposits" within the Doctoral Course "A.D.I." of Università degli Studi della Campania "L. Vanvitelli".

The research has been also funded by Università degli Studi della Campania 'L. Vanvitelli' through the programme "VALERE: VAnviteLli pEr la RicErca".

## Competing interests

At least one of the (co-)authors is a member of the editorial board of Hydrology and Earth System Sciences.

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
