# Peer review of "Understanding hydrologic controls of sloping soil response to precipitation through Machine Learning analysis applied to synthetic data"

_EGUsphere, 2022_

## Referee Comment (RC1)

Dear authors,

Thanks for your contribution on a hot topic regarding hydrology in mountainous and more generally sloping areas, which can indeed be of interest in many perspectives (risk management, agriculture/forestry, water supply).

**Summary of the preprint**

The goal of this study is to determine, amongst the parameters (rainfall, groundwater water head, and two soil water content) of an already calibrated 1D physical model, which are the main drivers (in the form of the most important antecedent conditions) of the soil water content. Indeed, this variable has been identified as critical in the occurence of landslides.

To do so, the field data is enriched with synthetic data produced by the 1D physical model, fed by stochastically-generated rainfall events. This augmented data pool is analysed through a machine learning method, combining Random Forest to assess the variable importance of each antecedent parameter conditions with regard to the soil water content output, and K-means clustering to classify the results.

Provided that the output is normalized (ratio between the soil water content and the rainfall event depth), the analysis shows a quite balanced importance of the antecedents (with a slight predominance of the soil water content condition). The triplet {rainfall, water content of the first meter of soil, groundwater head} is the most predictive of the soil water content evolution.

Finally, the classification reveals seasonal behaviors consistent with previous studies and the field observations.

**General comments**

- The article is well written, at the exception of some phrasing and vocabulary issues. Especially, I reckon that "soil cover" is misused as it is a synonym of "land use" and refers to vegetation or anything that covers the soil. In the article, it is used as a synonym of "soil" or more precisely "topsoil". Please correct this, as the land cover is not a variable nor a parameter in your model. Also, "slope response" is a bit spurious phrasing as what you're investigating is the soil water content response in a sloping context, and not that of the slope itself. I would suggest to use soil instead of slope and maybe rethink the title accordingly to avoid confusion (e.g. Understanding hydrologic controls of sloping soils response to precipitation...).

- The structure is good and most of the sections (or sub-sections) are clear (the description of the site and the field surveys, the clustering analysis etc.).

- Nevertheless, the aim of the study should be more precisely explained at the beginning. Is it oriented toward a seasonal analysis (what it seems, regarding the Results section and some mentions beforehand) or a crisis-warning model (which is also mentioned both in the Introduction and Conclusion, and would be more consistent with the time resolution of the data)? In my opinion, it can't do both. Depending on

the objective, the design choices for the synthetic data (time resolution, separation criterion) would be quite different.

- A figure (a workflow for instance) could be of help to summarize the method.

- Some information on the hardware used and the computation time would be welcome, especially by comparing it with another approach (sensitivity analysis).

**Specific comments**

- The method is based on a physical model for simulate the variable relationships on one hand (calibration), and produce the synthetic data on another hand (datapool augmentation). Could you not reach the same goal (or at least compare your results) with a sensitivity analysis of the 1D model with respect to the initial conditions (as you're focusing on the antecedents). In that regard, some supplementary elements supporting the choice of the method would be welcome.

- Concerning the synthetic rainfall generation, I assume that the separation criterion has a huge impact on the outcome of your methodology, especially if you aim at identifying the dominant parameter during extreme events (when the risk is the highest). Is there not a contradiction in chosing a 24h separation criterion (i.e. the time for the topsoil to drain almost completely) when the goal of the study is to assess the importance of the prior state of the soil? Could you elaborate a bit on the choice of this separation criterion (is it only based on the physics of the phenomenon, or is it constrained by the number of events you can bring into the machine-learning and still be computationaly-reasonable)?

- Did you try to assess the sensitivity of your method to the value of this separation criterion?

- On a related matter, by producing very diverse events in terms of duration and intensity, are you not risking to blur (to average) the relative importance of each parameter that might differ depending of the type of event? Said otherwise, if the goal is to find the sensitivity of your model to the prior conditions when an extreme event occurs, why not choose only extreme events for the analysis? Or for seasonality, why not splitting the rainfall chronicles beforehand and therefore acquire more specific relative importance of the antecedents?

- I understood that the previous studies were focusing on analysing and predicting the seasonal changes. Now, in this study, is this time-scale still relevant? I thought that the hour-time resolution was aiming at refining a model and a monitoring network more crisis-oriented (informing on the risk of a landslide to occur for example).

- Line 54: please, explicit what you mean by "long timescales".

- Line 98: Would not it be "at the contact between soil and bedrock" as soil cover is the description of what covers the surface of the soil? (Same remark for all "soil cover" occurrences)

- Lines 98-104: That refers to epikarst. You should maybe cite supplementary studies and modelling approaches outside your workgroup (Perrin et aL, 2003; Hartmann et al., 2014; Dal Soglio et al., 2020 for instance).

- Line 117: "identified" appears a bit confusing here, and can be understood only once the Method

section has been read. I would suggest "sorted" or "chosen". I also suggest to rephrase the whole sentence, whose syntax seems wrong to me.

- Line 321: Note that a purely 1D approach does not account for flow accumulation and possible secondary infiltration (runoff that infiltrate during its course downstream). That could overestimate the influence of the groundwater level by underestimating the amount of infiltration.

- Lines 373-376 and 491-493: How do you support this direct relationship between the water level in the aquifer and the one in the stream? No dedicated parameter appears in the mathematical description of the model. Moreover, a direct proportionnality might not be true, especially during extreme events (droughts and floods).

- Once again, I think that the Conclusion and the contribution of this study would be more valued if it were compared to another approach (local sensitivity analysis for instance).

**Technical corrections**

- In text reference ordering (reference grouped in the same parenthesis) should follow a consistent pattern (chronologically I would suggest, see HESS editors to make sure).

- Replace "soil cover" by "soil" or "topsoil" all along the article if you're agreeing with my previous statement on the meaning of these terms.

- Replace "slope" by "sloping soil" or simply "soil" as needeed.

- Line 136: "is" is missing between "results" and "quite variable".

- Line 439: "developed" is not appropriate here. Use "performed" or "carried out".

- Line 477: is" is missing between "soil storage" and "less connected". Maybe you shoud rephrase this sentence.

- Figures 5 and 6: the unit for the h_a (groundwater level) is mm. Shouldn't it be m? What is the base level?

---

## Author Comment (AC1)

**Reply to Reviewer #1**

On behalf of all co-authors I would like to thank the Reviewer for the time implied in this extensive revision of our manuscript. Responding to every comment individually, you will find your comments in regular style and our response in italic as follows:

Dear authors,

Thanks for your contribution on a hot topic regarding hydrology in mountainous and more generally sloping areas, which can indeed be of interest in many perspectives (risk management, agriculture/forestry, water supply).

Summary of the preprint

The goal of this study is to determine, amongst the parameters (rainfall, groundwater water head, and two soil water content) of an already calibrated 1D physical model, which are the main drivers (in the form of the most important antecedent conditions) of the soil water content. Indeed, this variable has been identified as critical in the occurence of landslides.

To do so, the field data is enriched with synthetic data produced by the 1D physical model, fed by stochastically-generated rainfall events. This augmented data pool is analysed through a machine learning method, combining Random Forest to assess the variable importance of each antecedent parameter conditions with regard to the soil water content output, and K-means clustering to classify the results.

Provided that the output is normalized (ratio between the soil water content and the rainfall event depth), the analysis shows a quite balanced importance of the antecedents (with a slight predominance of the soil water content condition). The triplet {rainfall, water content of the first meter of soil, groundwater head} is the most predictive of the soil water content evolution.

Finally, the classification reveals seasonal behaviors consistent with previous studies and the field observations.

*Thank you for so clearly summarizing the contents of our manuscript, which make us believe that, overall, the aim of the paper was clear enough and supported by the results.*

General comments

The article is well written, at the exception of some phrasing and vocabulary issues. Especially, I reckon that "soil cover" is misused as it is a synonym of "land use" and refers to vegetation or anything that covers the soil. In the article, it is used as a synonym of "soil" or more precisely "topsoil". Please correct this, as the land cover is not a variable nor a parameter in your model. Also, "slope response" is a bit spurious phrasing as what you're investigating is the soil water content response in a sloping context, and not that of the slope itself. I would suggest to use soil instead of slope and maybe rethink the title accordingly to avoid confusion (e.g. Understanding hydrologic controls of sloping soils response to precipitation...).

*Thank you for your positive evaluation of our work. Indeed, the term "soil cover" is sometimes used in landslide community, to define the mantle of soil (or regolith), that often covers a more compact and stable bedrock formation. We understand that the use of such term is misleading in the broader context of hillslope hydrology, so we will modify it to "soil mantle" throughout the text. We also accept your suggestion to modify*

*the title to "Understanding hydrologic controls of sloping soil response to precipitation through Machine Learning analysis applied to synthetic data".*

- The structure is good and most of the sections (or sub-sections) are clear (the description of the site and the field surveys, the clustering analysis etc.).

*Thank you again for your positive evaluation of the structure of the manuscript.*

- Nevertheless, the aim of the study should be more precisely explained at the beginning. Is it oriented toward a seasonal analysis (what it seems, regarding the Results section and some mentions beforehand) or a crisis-warning model (which is also mentioned both in the Introduction and Conclusion, and would be more consistent with the time resolution of the data)? In my opinion, it can't do both. Depending on the objective, the design choices for the synthetic data (time resolution, separation criterion) would be quite different.

*Indeed, the main aim of the study is to understand how the seasonal slope conditions, related to climate forcing, may affect the capability of the soil of retaining rainwater infiltration for a time long enough to potentially determine critical conditions as a consequence of rainfall events (e.g., the triggering of landslides). The time resolution of the data, as well as the criterion adopted to separate events within the rainfall record, are indeed tailored to this aim. In the revised manuscript, we will make it clearer in the final part of the Introduction.*

- A figure (a workflow for instance) could be of help to summarize the method.

*Thank you for suggesting it. We will consider adding a flowchart in the revised manuscript.*

- Some information on the hardware used and the computation time would be welcome, especially by comparing it with another approach (sensitivity analysis).

*In the revised manuscript, we will add some considerations about the computational effort. However, we do not think it is worth also adding a sensitivity analysis, which is out of the scope of our study, for the following reasons.*

*First, we analyzed the dataset mimicking what could be done if, rather than synthetically generated data, one was handling real field monitoring data. In fact, we were mostly looking for a way to identify the major cause-effect relationships between (measurable) inputs and outputs before (possibly, but not necessarily) building a model for the interpretation of such relationships, rather than evaluating the sensitivity of an (already available) model output to variations in the input (although the Random Forest analysis also allows quantifying the information content of each considered input variable).*

*Second, the sensitivity analysis is usually carried out to evaluate the effects of input (and parameter) uncertainty on model predictions. In this study, the model chain (already calibrated and validated previously: Greco et al., 2013; Comegna et al., 2016; Greco et al., 2018) is used as a tool to generate a (richer) synthetic dataset (this is a common problem in landslide studies, as field monitoring data records, even when they are relatively long, usually contain very few data representative of potentially critical situations). The model is assumed to represent "the reality", and adding a sensitivity analysis may result misleading, as it would move the focus to the performance of the model (which, in general, could also not exist).*

*Third, the adopted Random Forest analysis, which allows highlighting the most informative combination of measurable variables to predict the output, is somehow a sensitivity analysis as well, as it gives some indications about the relative importance of the input variables on the possibility of predicting the output, without introducing any mathematical model structure, but simply relying on the application of logical operators (IF-THEN-ELSE) between the variables.*

*In the revised manuscript, we will add paragraphs in the Introduction and in the Materials and Methods sections, to better explain the choice of Random Forest instead of a sensitivity analysis.*

Specific comments

- The method is based on a physical model for simulate the variable relationships on one hand (calibration), and produce the synthetic data on another hand (datapool augmentation). Could you not reach the same goal (or at least compare your results) with a sensitivity analysis of the 1D model with respect to the initial conditions (as you're focusing on the antecedents). In that regard, some supplementary elements supporting the choice of the method would be welcome.

*As already mentioned in our reply to one of the general comments, we are studying the relationships between the data as if they were measurements collected in the field, so without the recourse to any mathematical model (in our case, the model was just a tool to enrich the available dataset, so to make it significant for statistical analyses). In the revised manuscript, we will add paragraphs in the Introduction and in the Materials and Methods sections, to better explain the choice of Random Forest instead of a sensitivity analysis.*

- Concerning the synthetic rainfall generation, I assume that the separation criterion has a huge impact on the outcome of your methodology, especially if you aim at identifying the dominant parameter during extreme events (when the risk is the highest). Is there not a contradiction in chosing a 24h separation criterion (i.e. the time for the topsoil to drain almost completely) when the goal of the study is to assess the importance of the prior state of the soil? Could you elaborate a bit on the choice of this separation criterion (is it only based on the physics of the phenomenon, or is it constrained by the number of events you can bring into the machine-learning and still be computationaly-reasonable)?

*The separation of events within the continuous rainfall record aims at linking the occurrence (or non-occurrence) of critical conditions to a rainfall event, so that they can be considered as a direct consequence of that rainfall event. This is commonly made when empirical predictive tools (e.g., rainfall thresholds: Segoni et al., 2018a, b; Guzzetti et al., 2020; Piciullo et al., 2020) are implemented as part of early warning systems, e.g., against rainfall-induced landslides or debris flows, and the definition of the separation criterion is usually made empirically, looking at the performance of the predictor with different choices of the separation criterion.*

*From a physical viewpoint, especially when one is interested in the separation between the role of antecedent conditions, i.e., related to previous precipitation (and drainage/evapotranspiration) history, from the direct effects of the last precipitation event, it is quite complex to define a suitable separation criterion, especially if dealing with slow processes activated by precipitations, such as the infiltration through the unsaturated soil layer. In fact, to completely separate what depends on "previous" precipitation from what is linked to the last rainfall event, one should wait for the infiltration process initiated by previous precipitations to be finished, and, in a soil layer of few meters thickness, it may take several days. Extending so much the dry time interval between two separate events, especially during rainy seasons, would imply the aggregation of several events in a single one, thus leading to long rainy periods, rather than events, thus preventing the desired separation of antecedent conditions from direct effects of events. So, as we have defined the "response" of the soil layer as its attitude to retain infiltrated rainwater after the end of a rain event, looking at the moisture of the topsoil layer seemed a good trade-off: topsoil moisture controls the infiltration at the ground surface, hence when gravitational drainage from the topsoil is already over (the field capacity has been reached), the infiltration of a new rainfall input through the ground surface would not depend (or it would only little depend) on the remnants of the infiltration process caused by previous precipitation. In this respect, we tested a separation dry interval of 24 hours, commonly used when the available rainfall data are at daily resolution (Berti et al.,*

*2012; Leonarduzzi et al., 2017; Peres et al., 2018), and anyway in line with the empirical choices that are commonly made in the early warning community (Segoni et al., 2018a).*

*As we mentioned in the paper, the choice of 24 hours for the separation dry interval leads to about 50 rainfall events per year (i.e., 53061 rainfall events in 1000 years). The adopted Machine Learning techniques for the analysis of the dataset (K-means clustering and Random Forest algorithm) can handle larger datasets, thus the adoption of a shorter separation time interval, which would lead to a larger number of separated rainfall events, could be feasible from the computational effort point of view. However, we chose 24 hours for the previously explained reasons.*

- Did you try to assess the sensitivity of your method to the value of this separation criterion?

*No. We did not test different separation criteria.*

- On a related matter, by producing very diverse events in terms of duration and intensity, are you not risking to blur (to average) the relative importance of each parameter that might differ depending of the type of event? Said otherwise, if the goal is to find the sensitivity of your model to the prior conditions when an extreme event occurs, why not choose only extreme events for the analysis? Or for seasonality, why not splitting the rainfall chronicles beforehand and therefore acquire more specific relative importance of the antecedents?

*As already mentioned in our reply to previous comments, the goal of the study, which was not clearly described in the Introduction, is not to find the sensitivity of a model, but to find the most important cause-effect relationships between data, which could be a useful information to build a model. However, although extreme rainfall events are more likely leading to critical conditions in terms of increase of water storage in the soil, considering antecedent conditions may help not only to explain why extreme rainfall events sometimes do not lead to critical conditions, but also why sometimes ordinary (or not so extreme) rainfall events do cause critical conditions. About a-priori considering seasonality, our dataset clearly shows that, owing to climate variability, seasons are often anticipated or delayed, and the idea is that monitoring suitable variables may allow recognizing the actual establishment of "seasonal" conditions.*

- I understood that the previous studies were focusing on analysing and predicting the seasonal changes. Now, in this study, is this time-scale still relevant? I thought that the hour-time resolution was aiming at refining a model and a monitoring network more crisis-oriented (informing on the risk of a landslide to occur for example).

*As already mentioned in our reply to one of the previous comments, the main aim of the study is to understand how the seasonal slope conditions, related to climate forcing, may affect the capability of the soil of retaining rainwater infiltration for a time long enough to potentially determine critical conditions as a consequence of rainfall events (e.g., triggering of landslides). So, data at hourly resolution are required for the assessment of hazard in real time, while the assessment of antecedent conditions requires a longer timescale, and the relevant data might be also acquired at a coarser resolution (indeed, both soil moisture and groundwater level dynamics are much slower than rainfall). Hence, it could be possible to adopt different time resolutions for rainfall data (e.g., hourly) and for hydrological data (groundwater and soil moisture could be acquired at daily resolution). However, if one manages a monitoring network capable of hourly resolution, then the same dataset can be used for both short timescale predictions (hazard assessment) and long timescale processes (infiltration/drainage/evapotranspiration affecting antecedent conditions).*

- Line 54: please, explicit what you mean by "long timescales".

*We will modify the sentence, by writing "timescales of weeks or even months, much longer than the duration of rainfall events, typically ranging between some hours and few days".*

- Line 98: Would not it be "at the contact between soil and bedrock" as soil cover is the description of what covers the surface of the soil? (Same remark for all "soil cover" occurrences)

*The sentence will be modified to "Recent studies show that the response of the soil mantle to precipitation is affected by the wetness of the interface with the underlying bedrock, which controls the leakage of water from the soil to the fractured limestone".*

- Lines 98-104: That refers to epikarst. You should maybe cite supplementary studies and modelling approaches outside your workgroup (Perrin et aL, 2003; Hartmann et al., 2014; Dal Soglio et al., 2020 for instance).

*Agreed. We will specify that the uppermost weathered part of the bedrock is indeed the epikarst, and we will add the relevant suggested references.*

- Line 117: "identified" appears a bit confusing here, and can be understood only once the Method section has been read. I would suggest "sorted" or "chosen". I also suggest to rephrase the whole sentence, whose syntax seems wrong to me.

*In the revised manuscript, the sentence will be rewritten as: "After sorting the rainfall events within the 1000 years hourly timeseries, a dataset is built with the antecedent conditions one hour before the beginning of each rainfall event. It includes the previously listed variables plus the total rainfall event depth and the change in water stored in the soil cover at the end of each rainfall event."*

- Line 321: Note that a purely 1D approach does not account for flow accumulation and possible secondary infiltration (runoff that infiltrate during its course downstream). That could overestimate the influence of the groundwater level by underestimating the amount of infiltration.

*Obviously, heterogeneities of the soil mantle (either morphological, e.g., slope inclination, soil mantle thickness, or physical, e.g., soil layers with different hydraulic properties) may induce 3D effects in the flow processes. However, 3D effects are expected to be not particularly significant in the studied slopes, for several reasons. First, owing to the geometry of the slopes (i.e., hundreds of meters long with a soil mantle of few meters), the water potential gradients are such that significant deviations of the flow from the vertical direction (or, more precisely, from the direction orthogonal to ground surface) can occur only when the soil approaches saturation, so that capillarity gradients become small and gravitational gradient prevails (along a steeply inclined slope, in this condition the component of the gradient parallel to the slope becomes significant). In addition, the attainment of soil saturation is very unlikely, owing to the very high porosity (as high as 75%). Furthermore, the high inclination angles, in most slopes larger than 35°, imply that slope failure (landslide) would occur before soil attains saturation. Finally, the very high hydraulic conductivity (as high as 30 mm/h), together with the usually unsaturated soil conditions (soil capillary potential rarely overcomes -0,5 m: Cascini et al., 2014; Comegna et al., 2016; Napolitano et al., 2016), makes overland runoff very small, even*

*during the most intense rainfall events (Greco et al., 2018; Marino et al., 2020). In short, lateral redistribution of infiltration flow can be considered quite small in the studied slopes. In the revised manuscript, we will add more information about the characteristics of the studied slopes and soil (Section 2.1), and we will give some justification of the use of the simplified 1D model in Section 2.2.2.*

- Lines 373-376 and 491-493: How do you support this direct relationship between the water level in the aquifer and the one in the stream? No dedicated parameter appears in the mathematical description of the model. Moreover, a direct proportionality might not be true, especially during extreme events (droughts and floods).

*Indeed, in the description of how the epikarst aquifer is schematized in the model used to generate the synthetic data (lines 327-332), we have only written that it is modelled as a "linear reservoir, that releases water "as deep groundwater recharge and spring discharge". This conceptualization of the aquifer behavior implies that the streamflow (supplied by the springs) is proportional to the water level in the perched aquifer. We understand that, written in this way, it is not clear to the reader, so we will extend the explanation in the revised manuscript. Obviously, the assumption of a linear relationship linking aquifer water level and spring outflow is a simplification of the reality, and we agree that deviations from linearity are expected, especially in extreme conditions. However, synthetic groundwater level data are used only to separate "low" levels (clusters 1 and 3 of Figures 8, 9 and 10) from "high" (cluster 2 of Figures 8, 9 and 10) or "very high" levels (cluster 4 of Fig. 10), and the same could be made with stream level data, which is probably easier to be measured in the field, compared to the groundwater level in a temporary aquifer.*

- Once again, I think that the Conclusion and the contribution of this study would be more valued if it were compared to another approach (local sensitivity analysis for instance).

*We hope that, in the revised manuscript, the goal of our study, which did not result clear in the current version, will become clearer. We are not dealing with the development of a mathematical model of the behavior of the soil mantle of the studied slope, but we are rather analyzing field data (though synthetic) to understand the major cause-effect relationships between (measurable) variables. This analysis may be carried out in absence of any model, to interpret field data.*

Technical corrections

- In text reference ordering (reference grouped in the same parenthesis) should follow a consistent pattern (chronologically I would suggest, see HESS editors to make sure).

*Thank you for catching this inconsistency. In the revised manuscript, in all cases we will follow the chronological order.*

- Replace "soil cover" by "soil" or "topsoil" all along the article if you're agreeing with my previous statement on the meaning of these terms.

*We will use the word "soil mantle" in place of "soil cover".*

- Replace "slope" by "sloping soil" or simply "soil" as needed.

*We will follow this Reviewer's suggestion. The title of the revised manuscript will also be changed accordingly.*

- Line 136: "is" is missing between "results" and "quite variable".

*We will rewrite as "In these three areas, the thickness of the soil covers is quite variable".*

- Line 439: "developed" is not appropriate here. Use "performed" or "carried out".

*We will replace "developed" with "carried out".*

- Line 477: is" is missing between "soil storage" and "less connected". Maybe you shoud rephrase this sentence.

*We will rephrase the sentence as: "The importance of $h_a$ on the response of the soil mantle suggests that, in some conditions, the change in soil storage is affected by the capability of water exchange between the soil mantle and the underlying aquifer, as it will be discussed in the following sections".*

- Figures 5 and 6: the unit for the h_a (groundwater level) is mm. Shouldn't it be m? What is the base level?

*Thank you for suggesting. We will express $h_a$ in meters, as this unit is much more convenient for the groundwater level. The groundwater level is referred to the base of the epikarst, which is assumed 14 m below the interface between the soil mantle and the bedrock (Table 1). We will specify this in the revised manuscript.*

*References*

*Berti, M., Martina, M. L. V., Franceschini, S., Pignone, S., Simoni, A., & Pizziolo, M. (2012). Probabilistic rainfall thresholds for landslide occurrence using a Bayesian approach. Journal of Geophysical Research: Earth Surface, 117(4). https://doi.org/10.1029/2012JF002367*

*Cascini, L., Sorbino, G., Cuomo, S., & Ferlisi, S. (2014). Seasonal effects of rainfall on the shallow pyroclastic deposits of the Campania region (southern Italy). Landslides, 11(5), 779–792. https://doi.org/10.1007/s10346-013-0395-3*

*Comegna, L., Damiano, E., Greco, R., Guida, A., Olivares, L., & Picarelli, L. (2016). Field hydrological monitoring of a sloping shallow pyroclastic deposit. Canadian Geotechnical Journal, 53(7), 1125–1137. https://doi.org/10.1139/cgj-2015-0344*

*Dal Soglio, L.; Danquigny, C.; Mazzilli, N.; Emblanch, C.; Massonnat, G. Taking into Account both Explicit Conduits and the Unsaturated Zone in Karst Reservoir Hybrid Models: Impact on the Outlet Hydrograph. Water 2020, 12, 3221. https://doi.org/10.3390/w12113221*

*Greco, R., Comegna, L., Damiano, E., Guida, A., Olivares, L., & Picarelli, L. (2013). Hydrological modelling of a slope covered with shallow pyroclastic deposits from field monitoring data. Hydrology and Earth System Sciences, 17(10), 4001–4013. https://doi.org/10.5194/hess-17-4001-2013*

*Greco, R., Marino, P., Santonastaso, G. F., & Damiano, E. (2018). Interaction between perched epikarst aquifer and unsaturated soil cover in the initiation of shallow landslides in pyroclastic soils. Water (Switzerland), 10(7). https://doi.org/10.3390/w10070948*

Guzzetti, F., Gariano, S. L., Peruccacci, S., Brunetti, M. T., Marchesini, I., Rossi, M., & Melillo, M. (2020, January 1). *Geographical landslide early warning systems. Earth-Science Reviews.* Elsevier B.V. https://doi.org/10.1016/j.earscirev.2019.102973

Hartmann, A., Goldscheider, N., Wagener, T., Lange, J., & Weiler, M. (2014, September 1). *Karst water resources in a changing world: Review of hydrological modeling approaches. Reviews of Geophysics.* https://doi.org/10.1002/2013RG000443

Leonarduzzi, E., Molnar, P., & McArdell, B. W. (2017). *Predictive performance of rainfall thresholds for shallow landslides in Switzerland from gridded daily data. Water Resources Research, 53(8),* 6612–6625. https://doi.org/10.1002/2017WR021044

Marino, P., Comegna, L., Damiano, E., Olivares, L., & Greco, R. (2020). *Monitoring the hydrological balance of a landslide-prone slope covered by pyroclastic deposits over limestone fractured bedrock. Water (Switzerland), 12(12).* https://doi.org/10.3390/w12123309

Napolitano, E., Fusco, F., Baum, R. L., Godt, J. W., & De Vita, P. (2016). *Effect of antecedent-hydrological conditions on rainfall triggering of debris flows in ash-fall pyroclastic mantled slopes of Campania (southern Italy). Landslides, 13(5),* 967–983. https://doi.org/10.1007/s10346-015-0647-5

Peres, D. J., Cancelliere, A., Greco, R., & Bogaard, T. A. (2018). *Influence of uncertain identification of triggering rainfall on the assessment of landslide early warning thresholds. Natural Hazards and Earth System Sciences, 18(2),* 633–646. https://doi.org/10.5194/nhess-18-633-2018

Perrin, J., Jeannin, P. Y., & Zwahlen, F. (2003). *Epikarst storage in a karst aquifer: A conceptual model based on isotopic data, Milandre test site, Switzerland. Journal of Hydrology, 279(1–4),* 106–124. https://doi.org/10.1016/S0022-1694(03)00171-9

Piciullo, L., Tiranti, D., Pecoraro, G., Cepeda, J. M., & Calvello, M. (2020). *Standards for the performance assessment of territorial landslide early warning systems. Landslides, 17(11),* 2533–2546. https://doi.org/10.1007/s10346-020-01486-4

Segoni, S., Piciullo, L., & Gariano, S. L. (2018a). *A review of the recent literature on rainfall thresholds for landslide occurrence. Landslides.* Springer Verlag. https://doi.org/10.1007/s10346-018-0966-4

Segoni, S., Rosi, A., Fanti, R., Gallucci, A., Monni, A., & Casagli, N. (2018b). *A regional-scale landslide warning system based on 20 years of operational experience. Water (Switzerland), 10(10).* https://doi.org/10.3390/w10101297

---

## Author Comment (AC2)

**Reply to Reviewer #2**

This paper aims at understanding the responses of slopes to precipitation. To do so the authors rely on mathematical and machine learning models that have been developed by using synthetic data inspired by ground measurements.

This is an interesting topic and the authors have performed a significant number of experiments to shed light on the responses of slopes (although I don't think these experiments account for the effects of slope) to precipitation in mountainous regions.

The organization of the paper is clear however, the English language needs to be improved.

*Thank you for your positive evaluation of the manuscript. We agree that it is more appropriate to write "soil response", and we will change it throughout the revised paper, as well as in the title, which will be modified in "Understanding hydrologic controls of sloping soil response to precipitation through Machine Learning analysis applied to synthetic data". We will double-check the English language, so to fix all grammar and spelling mistakes.*

Below are my comments:

The title of the paper is "slope responses to precipitation" but actually only a 1-D simplified model has been performed therefore, no slopes have been modelled. For such a study, I would expect a 2-D or even a 3-D model with the slope and lateral flow. This is a big limitation of the study as accounting for slopes will change the results presented in this paper.

*As already mentioned in the reply to the previous comment, the title of the paper will be modified to: "Understanding hydrologic controls of sloping soil response to precipitation through Machine Learning analysis applied to synthetic data". However, we would like to make some remarks about the supposed big limitation due to the choice of a simplified 1D model to mimic the field data. Obviously, heterogeneities of the soil mantle (either morphological, e.g., slope inclination, soil mantle thickness, or physical, e.g., soil layers with different hydraulic properties) may induce 3D effects in the flow processes. However, 3D effects are expected to be not particularly significant in the unsaturated soil mantle of the studied slopes, for several reasons. First, owing to the geometry of the slopes (i.e., hundreds of meters long with a soil mantle of few meters), the water potential gradients are such that significant deviations of the flow from the vertical direction (or, more precisely, from the direction orthogonal to ground surface) can occur only when the soil approaches saturation, so that capillarity gradients become small and gravitational gradient prevails (along a steeply inclined slope, in this condition the component of the gradient parallel to the slope becomes significant). In addition, the attainment of soil saturation is very unlikely in the studied soil, owing to its very high porosity (as high as 75%). Furthermore, the high inclination angles, in most slopes larger than 35°, imply that slope failure (landslide) would occur before soil attains saturation. Finally, the very high hydraulic conductivity (as high as more than 30 mm/h), together with the usually unsaturated soil conditions (soil capillary potential rarely overcomes -0,5 m: Cascini et al., 2014; Comegna et al., 2016; Napolitano et al., 2016), makes overland runoff very small, even during the most intense rainfall events (Greco et al., 2018; Marino et al., 2020). In short, lateral redistribution of infiltration flow can be considered quite small in the soil mantle of the studied slopes. In the revised manuscript, we will add more information about the characteristics of the studied slopes and soil (Section 2.1), and we will give some justification of the use of the simplified 1D model in Section 2.2.2.*

I am not familiar with the term "soil cover" what does it exactly mean?

*Soil cover is used in the landslide research literature to define a mantle of soil, of various origins, covering a more compact and stable bedrock. We understand that this is misleading for the broader readership of hillslope hydrologists, so we will change it to "soil mantle" throughout the entire revised manuscript.*

Write "Precipitation" without and s

*Thank you for catching this mistake. We will make the correction throughout the entire manuscript, including the title.*

The abstract is quite long, shorten it.

*Indeed, the abstract is currently more than 2000 characters. Although the guidelines for article preparation do not fix limits, it is far too long. In the revised manuscript, we will completely rewrite it. We have already drafted it, and it looks that there is room to reduce it to about 60% of its current length.*

L17-26: from the paper, most of the work was based on synthetic data.

*In the revision of the abstract, we will clarify that the study deals with the synthetic dataset.*

One of the objectives of this paper was to find out the variables to be measured. This objective has not been addressed. It is also obvious that soil moisture, groundwater levels, and rainfall are the variables that should be monitored.

*Field monitoring for the assessment of rainfall-induced geohazards usually involves only rainfall measurements (Peruccacci et al., 2017). Only recently the importance of soil moisture measurements for the prediction of shallow landslides and debris flows is being recognized (Lazzari et al., 2018; Mirus et al., 2018; Marino et al., 2020). About groundwater level, it is usually considered an informative variable only for deep-seated landslides, as in that case the slip surface of the landslide can be below the groundwater table. Differently, to the best of our knowledge, it has been never recognized useful also for shallow landslide prediction. Hence, in the context of our study, monitoring it is far from being obvious.*

*We understand that, in the Introduction, the focus on geohazards, and specifically on shallow landslides and debris flows was not clearly stated (see also the following comment from this Reviewer). We will modify the final part of the Introduction to better describe the context and the goal of the study.*

The paper did not link landslides to slopes. The authors should clearly state that their work aims to identify the parameters controlling the responses of slopes to precipitation with implications for landslides since the landslides have not been specifically addressed.

*We agree. We will rewrite parts of the Introduction to state more clearly that the response of the soil mantle to precipitation is studied for rainfall-induced geohazard assessment and, more specifically, for shallow landslide and debris flow hazard assessment. Also, in the discussion of the results we will add some text to explicitly state the implications that the different conditions may have for slope stability assessment.*

L56: "has been early identified" reword.

*We will rephrase the entire sentence between lines 55 and 58, which will become: "While the importance of soil moisture conditions on slope runoff and drainage has been recognized long since (Ponce & Hawkins, 1996; Tromp-Van Meerveld & McDonnell, 2006a, 2006b), …"*

L83-84: "but where particularly destructive rainfall triggered landslides occurred." Reword

*We will rephrase lines 83-85 as: "This research focuses on a case study in an area frequently hit by destructive rainfall-triggered shallow landslides".*

L94 avoid starting a sentence with "not only". "Not only" then what?

*We will rephrase lines 95-99 as: "Recent studies show that the response of the soil mantle to precipitation is affected not only by rainfall characteristics and antecedent soil moisture, but also by the wetness of the soil-bedrock interface, which controls the leakage of water into the underlying fractured limestone (Marino et al., 2020a, b).*

L112: change to 1000-year without the s or remove the hyphen.

*Thank you for the suggestion. We will use 1000 years throughout the entire manuscript.*

L149: remove a priori.

*Agree. "A priori" is not necessary and will be removed.*

L220: change to November 11th 2021

*Thank you. We will delete the word "the".*

L260: is the synthetic rainfall close to reality? Is the rainfall consistent with the climatology of the area. Could you show the comparisons between real rainfall data and the ones you have created? Are the wet and dry interval consistent with reality? Could you provide some comparisons with the real-world data?

*Thank you for raising these issues. In the submitted manuscript, we decided to describe very briefly the stochastic NRSP model used for synthetic rainfall generation, giving some references to let the interested readers get more information. We understand that we gave too little information, given that the synthetic rainfall series plays an important role in our methodology. In the following, we give detailed information to the Reviewer, so that he can judge about how the generated synthetic rainfall resembles the real experimental record. In the revised manuscript, we will try to find a trade off between the sake of brevity (the synthetic rainfall generation is here only a tool, but it is not the core of the study) and the need for more information. Possibly we will put some of the information in an appendix.*

*The NRSP stochastic model of rainfall (Neyman and Scott, 1958; Rodriguez-Iturbe et al., 1987a, b; Cowpertwait et al., 1996) describes the process of point rainfall as a superposition of randomly arriving rain clusters, each containing several rain cells with constant intensity. The hyetograph within a cluster is obtained by the superposition of the intensity of the various cells belonging to the cluster. It has been calibrated based on 17 years experimental data (2000-2016) of rainfall depth at 10 min resolution, recorded by the rain gauge managed by Civil Protection in Cervinara. The calibration has been carried out by minimizing, for rainfall aggregated at various durations, the difference between the following quantities, estimated by the model and calculate from the experimental data: mean, variance, lag 1 autocorrelation, probability of dry interval,*

probability of transition from dry-to-dry interval, probability of transition from wet-to-wet interval. The calibration procedure is based on the one proposed by Coptwertwait et al. (1996), and it is described in detail in Peres and Cancelliere (2014). To account for the seasonality of rainfall, these quantities have been calculated month by month in the experimental record (Fig. R1), suggesting that the calibration of the NRSP model should be carried out separately for seven homogeneous periods (September, October, November, December-March, April, May-June, July-August).

[revised manuscript text omitted]

Did you use a specific code for the hydrologic modeling, if yes please provide reference to the code, if not add the detail about the code you use.

*We solved the integration of the 1D Richards' equation, with the conditions assigned at the upper boundary and at the interface with the bedrock, coupled with the continuity equation of the linear reservoir schematizing the perched aquifer in the epikarst, with a self-made finite difference code written in Matlab environment. We believe that adding too many details about the numerical solver in the paper is not necessary, as it is a standard numerical technique, and it would further increase the length of an already long paper.*

Even if these test cases are synthetic, you should compare the results to some observations to check if it fits within the boundaries of the variables. For example, soil moisture and groundwater levels can be compared to ground measurements.

*The comparison with field data is indeed what we meant to do with Figure 5 (and Figure 6). From those figures, you can directly compare the few measured values of soil moisture of the upper 100 cm of the soil profile with the synthetic data. About the synthetic groundwater level data, they are compared with stream water level data. The reasons for this choice are several:*

*- So far, we have measurements of stream water level, while only recently we have installed two piezometers in the epikarst.*

*- The streams are supplied by groundwater coming from the fractured bedrock with very little contribution of overland runoff (less than 1% of the rainfall) only during the most intense rainstorms (it is revealed by the timing of the observed hydrographs in response to rainfall as well as by measurements of electric conductivity of stream water: Marino et al., 2020), so there might be a close relationship linking stream water level and groundwater level.*

*- Installing piezometers in the fractured limestone is a complex operation, owing to the mechanical resistance of the rock, which obliges to the use of powerful drilling machines; we have recently installed two piezometers (July 2020), but one of them could penetrate the limestone only for less than a couple of meters, as the machine that could be carried in that steep part of the slope (a light one) was not able to drill more depth; the second piezometer, which is at the foot of the slope, in a much less steep terrain, penetrates 16 meters below the ground, but there we have found a different kind of soil mantle (not only pyroclastic soil, but also some meters of alluvial deposits), in total more than 10 meters thick; as we had no clue of the degree of interconnection of the fractured system in the limestone, we decided to extend the pervious part of the piezometer (the filter) to almost the entire penetration depth in the limestone (1,5 meters for the first piezometer, 5 meters for the second one), as a shorter filter at the base of the piezometer (as it is usually done) would increase the risk of not intercepting any connected fracture; in this way, there is more chance for water to enter the piezometer, but, as it may enter at any height along the filter and then pond at the base of the piezometer, we cannot convert the water depth that we measure in the piezometer into a groundwater level; during the 2020/2021 hydrologic year we did not measure any water in the piezometers (2020 was a quite dry year), but in December 2021, after a quite rainy autumn (more than 900 mm between September and December), for the first time water appeared in both the piezometers, confirming that the temporary aquifer actually develops in the epikarst during rainy periods; until summer 2022, the piezometric measurements were made irregularly with a freatimeter, but in autumn 2022 we have installed an automatic sensor inside the piezometer on the steep terrain (the first one), and this winter we have observed a slight increase of groundwater level once the cumulated rainfall from September exceeded 800 mm.*

*- Stream water seems to appear and disappear consistently with the groundwater fluctuations, although, so far, we have too few data to demonstrate it; however, measuring stream water level is much easier than groundwater level in the studied context, and it could be an effective surrogate of groundwater level.*

*- The use that we do with the synthetic groundwater level data (that could be done with field data, either of groundwater or of stream water level) is just to discriminate between "high" level and "low" level, as a proxy to identify active subsurface drainage conditions.*

*The colored dots of Figures 5 and 6 also show that the seasonality of the synthetic variables is consistent with that of the observed variables.*

To understand the effects of each variable on the hydrologic processes, a sensitivity analysis could be performed. Why did the authors choose machine learning technique? I understand that the clustering needs a machine learning technique but to clearly highlight the importance of a variable to a given hydrologic processes, a sensitivity analysis could be performed.

*This comment, as well as similar comments made by the other Reviewers, clearly indicates that, in the Introduction, we failed to describe the aims of the study. In the revised manuscript, we will add paragraphs in the Introduction and in the Materials and Methods sections, to better explain the choice of Machine Learning (and specifically Random Forest) instead of a sensitivity analysis. In fact, we believe that adding a sensitivity analysis, which is out of the scope of our study, would be misleading, for the following reasons.*

*First, we analyzed the dataset mimicking what could be done if, rather than synthetically generated data, one was handling real field monitoring data. In fact, we were mostly looking for a way to identify the major cause-effect relationships between (measurable) inputs and outputs before (possibly, but not necessarily) building a model for the interpretation of such relationships, rather than evaluating the sensitivity of an (already available) model output to variations in the input (although the Random Forest analysis also allows quantifying the information content of each considered input variable).*

*Second, the sensitivity analysis is usually carried out to evaluate the effects of input (and parameter) uncertainty on model predictions. In this study, the model chain (already calibrated and validated previously: Greco et al., 2013; Comegna et al., 2016; Greco et al., 2018) is used as a tool to generate a (richer) synthetic dataset (this is a common problem in landslide studies, as field monitoring data records, even when they are relatively long, usually contain very few data representative of potentially critical situations). The model is assumed to represent "the reality", and adding a sensitivity analysis may result misleading, as it would move the focus to the performance of the model (which, in general, could also not exist).*

*Third, the adopted Random Forest analysis, which allows highlighting the most informative combination of measurable variables to predict the output, is somehow a sensitivity analysis as well, as it gives some indications about the relative importance of the input variables on the possibility of predicting the output, without introducing any mathematical model structure, but simply relying on the application of logical operators (IF-THEN-ELSE) between the variables.*

L655: what does it mean "monitoring antecedent conditions"

*The first paragraphs of the Conclusions section describe what we mean with "antecedent conditions": the values of mean soil moisture of the uppermost 100 cm of the soil mantle ($\vartheta_{100}$) and the water level in the perched aquifer stored in the epikarst ($h_a$) before the onset of each rainfall event. In the revised version of the manuscript, we will specify again what are the variables that the results, obtained with the analysis of the*

*synthetic dataset, suggest being useful, if monitored in the field, to predict the soil attitude to retain infiltrating rainwater.*

Figure 7: change to a and b instead of left and right. Same for Figure 9

*We will fix it in the revised manuscript.*

---

## Author Comment (AC3)

**Reply to Reviewer #3**

General comments

The manuscript describes a field study of a slope, enriched by simulation results from a 1D soil column model. There is relevance in the author's work and presented data, however, unfortunately, I struggle to find the novelty and contribution to current research in this work. All in all, it seems more like an analysis of the behaviour of a simple 1D-model (without considering the model's parameters), and it does not become clear what added value this research offers. I would not count monitoring antecedent conditions as new insight (conclusions, line 655)?

*It is clear from the comments of the Reviewer (as well as from those of the other Reviewers), that the way we wrote the Introduction, especially the description of the goal of the study, resulted unclear to the reader. The main aim of the study is to understand how the seasonal slope conditions, related to climate forcing, may affect the capability of the soil of retaining rainwater infiltration for a time long enough to potentially determine critical conditions as a consequence of rainfall events (e.g., the triggering of landslides).*

*To this aim, to enrich the available field dataset, in which very few times the soil mantle approached critical conditions (as it often occurs in studies dealing with geohazard assessment, as the hazardous conditions are rare per definition), we generated a synthetic dataset with already existing models, calibrated and validated in previous studies based on experimental data collected in the field (Greco et al., 2013; Comegna et al., 2016; Greco et al., 2018), but here the model is just a tool to generate the dataset. Once generated, the dataset represents "the reality", and we analyzed it mimicking what could be done if, rather than synthetically generated data, one was handling real field monitoring data. In fact, we were mostly looking for a way to identify the major cause-effect relationships between (measurable) inputs and outputs before (possibly, but not necessarily) building a model for the interpretation of such relationships, rather than evaluating the sensitivity of an (already available) model output to variations in the input (although the Random Forest analysis also allows quantifying the information content of each considered input variable, but without introducing any mathematical model structure, as it is based on the application of logical rules (IF-THEN-ELSE) to classify the input variables).*

*This is the reason why we did not test the effects of uncertainty/variability of model parameters: it was simply out of the scope of our study. The obtained results show, for the assessment of the hazard of rainfall-induced landslides and debris flows, the potential value of supplementing the monitoring of rainfall (which is the only monitored variable in nearly all the real world applications) with the monitoring of soil moisture before the rainfall (this is something that is recently being recognized by many researchers, but rarely adopted in operational hazard assessment systems), and the monitoring of the water level in the shallow aquifer developing in the uppermost part of the underlying bedrock. This is quite a novel result, to our knowledge never proposed to predict the response to precipitation of a shallow unsaturated soil mantle. Groundwater level is usually considered an informative variable only for deep-seated landslides, as in that case the (deep) slip surface of the landslide can be below the groundwater table. In the studied geomorphological context, monitoring these antecedent conditions is indeed a novelty.*

*In the revised manuscript, besides rewriting the Introduction, we will more clearly underline this novel aspect in the discussion of the results, as well as in the Conclusions.*

The 1D model comes necessarily with many simplifciations (which of course also has its advantages). But the shortcomints are not adressed (and the advantages - fast runtime etc. - not really exploited). E.g. heterogeneity of hydraulic conductivities on a slope, differences in lower aquifer pressure based on position (top, bottom, local gradient) in the slope, parameter uncertainty in general, differences in layer thicknesses across the slope, etc.

*We thank the Reviewer for raising this issue. Obviously, heterogeneities of the soil mantle (either morphological, e.g., slope inclination, soil mantle thickness, or physical, e.g., soil layers with different hydraulic properties) may induce 3D effects in the flow processes. However, 3D effects are expected to be not particularly significant in the soil mantle of the studied slopes, for several reasons. First, owing to the geometry of the slopes (i.e., hundreds of meters long with a soil mantle of few meters), the water potential gradients are such that significant deviations of the flow from the vertical direction (or, more precisely, from the direction orthogonal to ground surface) can occur only when the soil approaches saturation, so that capillarity gradients become small and gravitational gradient prevails (along a steeply inclined slope, in this condition the component of the gradient parallel to the slope becomes significant). In addition, the attainment of soil saturation is very unlikely, owing to the very high porosity (as high as 75%). Furthermore, the high inclination angles, in most slopes larger than 35°, imply that slope failure (landslide) would occur before soil attains saturation. Finally, the very high hydraulic conductivity (as high as 30 mm/h), together with the usually unsaturated soil conditions (soil capillary potential rarely overcomes -0.5 m: Cascini et al., 2014; Comegna et al., 2016; Napolitano et al., 2016), makes overland runoff very small, even during the most intense rainfall events (Greco et al., 2018; Marino et al., 2020). In short, lateral redistribution of infiltration flow can be considered quite small in the soil mantle of the studied slopes. In the revised manuscript, we will add more information about the characteristics of the studied slopes and soil (Section 2.1), and we will give some justification of the use of the simplified 1D model in Section 2.2.2.*

*About the variability of the groundwater table depth, this is also obviously true (and indeed, observations made in two piezometers, recently installed at two different altitudes along the slope, confirm that the groundwater table depth may be quite different). However, the use that we make of the groundwater level information is to discriminate "low" levels (clusters 1 and 3 of Figures 8, 9 and 10) from "high" levels (cluster 2 of Figures 8, 9 and 10) or "very high" levels (cluster 4 of Fig. 10). Depending on the availability of monitoring instruments, this could be made with a single piezometer, as well as with several piezometers (but, although with different levels, if the groundwater level in a piezometer is high, it will be likely high also in the others, unless they are so far from each other that they are monitoring disconnected groundwater systems). This aspect will be better clarified in the discussion of the results of the revised manuscript.*

Given such a relatively simple model, it should be possible to run uncertainty analysis or sensitivity analysis – and that possibility should be exploited.

Then, it is unclear why a Random Forest algorithm is used to emulate the physically-based model outputs – why not simply use an ensemble of the physically-based model, and do some uncertainty and sensitivity analysis on that? Also, some basic considerations when using RF have been ignored (hyperparameter search, sound cross validation strategy, discussion of size of dataset etc).

Therefore, I recommend to address the analysis of the physically-based model more extensively before venturing into a ML analysis of its results.

*This comments again clearly indicate that, in the Introduction, we failed to describe the aims of the study. In the revised manuscript, we will add paragraphs in the Introduction and in the Materials and Methods sections, to better explain the choice of Machine Learning (and specifically Random Forest) instead of a sensitivity analysis. In fact, we believe that adding a sensitivity analysis, which is out of the scope of our study, would be misleading, for the following reasons:*

*First, we analyzed the dataset mimicking what could be done if, rather than synthetically generated data, one was handling real field monitoring data. In fact, we were mostly looking for a way to identify the major cause-effect relationships between (measurable) inputs and outputs before (possibly, but not necessarily) building a model for the interpretation of such relationships, rather than evaluating the sensitivity of an (already available) model output to variations in the input (although the Random Forest analysis also allows quantifying the information content of each considered input variable).*

*Second, the sensitivity analysis is usually carried out to evaluate the effects of input (and parameter) uncertainty on model predictions. In this study, the model chain (already calibrated and validated previously: Greco et al., 2013; Comegna et al., 2016; Greco et al., 2018) is used as a tool to generate a (richer) synthetic dataset (this is a common problem in landslide studies, as field monitoring data records, even when they are relatively long, usually contain very few data representative of potentially critical situations). The model is assumed to represent "the reality", and adding a sensitivity analysis may result misleading, as it would move the focus to the performance of the model (which, in general, could also not exist).*

*Third, the adopted Random Forest analysis, which allows highlighting the most informative combination of measurable variables to predict the output, is somehow a sensitivity analysis as well, as it gives some indications about the relative importance of the input variables on the possibility of predicting the output, without introducing any mathematical model structure, but simply relying on the application of logical operators (IF-THEN-ELSE) between the variables.*

Specific comments

Section 1. Introduction

Line 94 ff: Remember that you are describing location-specific aspects – "fractured limestone", depth of soil above bedrock – i.e. this is not generally applicable.

*Thank you for catching this. Indeed, this paragraph refers specifically to the presented case study (i.e., slopes with a shallow pyroclastic soil mantle covering a fractured limestone bedrock). Therefore, it should be moved to the final part of the Introduction, where the characteristics of the case study are briefly anticipated.*

Line 105ff: This paragraph with your objectives could be formulated more clearly. E.g. the first sentence – please reformulate and state more clearly what your objectives are.

*Apart from some sentences, which should be rewritten to improve the language and style, this paragraph must be totally rewritten, as it misled the Reviewers (and it would mislead all the readers of the paper). In fact, as already pointed out in the replies to previous comments from this Reviewer, the focus of the study is not on the physically based model (which is instead, by our mistake, mentioned firstly in the paragraph of the objectives of the study), but on the interpretation of field monitoring data. Coupled with the NRSP stochastic model of rainfall, the model was just a tool to generate a rich synthetic dataset, which was then analyzed as if it were obtained by field measurements. This approach was chosen because field data series always contain few data representative of potentially critical conditions (in other words, landslides and debris flows, as well as other rainfall-induced geohazards are rare phenomena), so to have enough data to carry out statistically significant analyses. The final part of the Introduction will be completely reformulated to state the goal of the study more clearly.*

Section 2. Material and methods

Line 140, line 146: Make clear that you describe/summarize the methods applied in *your* study (e.g. by adding "see section 2.2")

*In the revised manuscript we will make more clear that this initial part of section 2 anticipates what is then described in the following subsections.*

Figure 1: I suggest a visualizing a DEM as background in one of the images – maybe the smaller inset? However, it remains unclear whether the inset is necessary at all, or one map would do just fine. Indicate the location of the monitoring station. Moreover, the outline of the inset seems incorrect, as well as the red star indicating the main scarp does not match the location indicated in the inset.

*We will use a DEM instead of a photo in the smaller inset, so to give the reader information about the morphology of the studied slope. We will also be more precise in selecting the zoomed rectangle, as well as we will move the red star in the correct position (thank you for catching this mistake), which should be the main scarp of the landslide.*

Figure 2: This is based on data from Damiano et al. 2012?

*Indeed Figure 2 is adapted from Damiano et al. (2012), and this should have been mentioned in the caption. However, as the layered nature of the soil cover is an information that is never exploited in this study, we are considering summarizing its description in the revised manuscript, likely eliminating the figure.*

Line 196: By "pyroclastic ashes" you here refer to the entire soil profile? Or only to the layer "Volcanic ashes" in Fig. 2? The terminology used around here should be made more clear.

*We agree that we should be consistent, using the word "pyroclastic" throughout the entire manuscript.*

Figure 3: Please improve this figure. E.g. north up, show its outline in Figure 1.

*In the revised manuscript, we will change the photo, using one with the standard orientation (North upward).*

Section 2.1.1 partly lacks details – what has been measured exactly? How long? What temporal resolution? (part of it comes later in section 3.2, and should be noted here)

*Thank you for the suggestion. In the revised manuscript we will add more information about the monitoring campaign, and we will also move to Section 2.1.1 the information given between lines 487 and 492 (Section 3.2), leaving there only a small mention.*

Section 2.2.1: Reference for the NSRP model is lacking. Also, some kind of comparison (various statistics?) of the synthetic time series with the observed time series would be appreciated.

*Thank you for raising these issues. In the submitted manuscript, we decided to describe very briefly the stochastic NRSP model used for synthetic rainfall generation, giving some references to let the interested readers get more information. We understand that we gave too little information, given that the synthetic rainfall series plays an important role in our methodology. In the following, we give detailed information to this Reviewer, so that he can judge about how the generated synthetic rainfall resembles the real experimental record. In the revised manuscript, we will try to find a trade off between the sake of brevity (the synthetic rainfall generation is here only a tool, but it is not the core of the study) and the need for more information. Possibly we will put some of the information in an appendix.*

The NRSP stochastic model of rainfall (Neyman and Scott, 1958; Rodriguez-Iturbe et al., 1987a, b; Cowpertwait et al., 1996) describes the process of point rainfall as a superposition of randomly arriving rain clusters, each containing several rain cells with constant intensity. The hyetograph within a cluster is obtained by the superposition of the intensity of the various cells belonging to the cluster. It has been calibrated based on 17 years experimental data (2000-2016) of rainfall depth at 10 min resolution, recorded by the rain gauge managed by Civil Protection in Cervinara. The calibration has been carried out by minimizing, for rainfall aggregated at various durations, the difference between the following quantities, estimated by the model and calculate from the experimental data: mean, variance, lag 1 autocorrelation, probability of dry interval, probability of transition from dry-to-dry interval, probability of transition from wet-to-wet interval. The calibration procedure is based on the one proposed by Coptwertwait et al. (1996), and it is described in detail in Peres and Cancelliere (2014). To account for the seasonality of rainfall, these quantities have been calculated month by month in the experimental record (Fig. R1), suggesting that the calibration of the NRSP model should be carried out separately for seven homogeneous periods (September, October, November, December-March, April, May-June, July-August).

[revised manuscript text omitted]

Line 284ff: For your definition of separate rainfall events – can you argue for the threshold of 24h with less than 2mm rainfall to separate events? This seems very little – does the slope really drain in such a short time? I.e. have effects of preceding events really disappeared after such a short time? I do not follow your argument that the volumetric water content at 10cm depth is sufficient to conclude that. Also, you show model results in Figure 4, right? That is not directly apparent to the reader.

*The separation of events within the continuous rainfall record aims at linking the occurrence (or non-occurrence) of critical conditions to a rainfall event, so that they can be considered as a direct consequence of that rainfall event. This is commonly made when empirical predictive tools (e.g., rainfall thresholds: Segoni et al., 2018a, b; Guzzetti et al., 2020; Piciullo et al., 2020) are implemented as part of early warning systems, e.g., against rainfall-induced landslides or debris flows, and the definition of the separation criterion is usually made empirically, looking at the performance of the predictor with different choices of the separation criterion.*

*From a physical viewpoint, especially when one is interested in the separation between the role of antecedent conditions, i.e., related to previous precipitation (and drainage/evapotranspiration) history, from the direct effects of the last precipitation event, it is quite complex to define a suitable separation criterion, specifically if dealing with slow processes activated by precipitations, such as the infiltration through the unsaturated soil layer. In fact, to completely separate what depends on "previous" precipitation from what is linked to the last rainfall event, one should wait for the infiltration process initiated by previous precipitations to be finished, and, in a soil layer of few meters thickness, it may take several days. Extending so much the dry time interval between two separate events, especially during rainy seasons, would imply the aggregation of several events in a single one, thus leading to long rainy periods, rather than events, thus preventing the desired separation of antecedent conditions from direct effects of events. So, as we have defined the "response" of the soil layer as its attitude to retain infiltrated rainwater after the end of a rain event, looking at the moisture of the topsoil layer seemed a good trade-off: topsoil moisture controls the infiltration at the ground surface, hence when*

*gravitational drainage from the topsoil is already over (the field capacity has been reached), the infiltration of a new rainfall input through the ground surface would not depend (or it would only little depend) on the remnants of the infiltration process caused by previous precipitation. In this respect, we tested a separation dry interval of 24 hours, commonly used when the available rainfall data are at daily resolution (Berti et al., 2012; Leonarduzzi et al., 2017; Peres et al., 2018), and anyway in line with the empirical choices that are commonly made in the early warning community (Segoni et al., 2018a).*

*We will clarify that Figure 4 shows synthetic data.*

Section 2.2.2: I miss a discussion of the limitations of the 1D model. Assuming that some lateral in the soil layer (and deeper aquifer) exists, this results in different groundwater level across different parts of the slope / for different gradients. Etc… You set up a single model (with a single parameter set) – ignoring the heterogeneity of soil thickness, hydraulic conductivites, etc. one would find along the slope? Given such a simple 1D model, I would at least recommend to perform some kind of sensitivity analysis / parameter uncertainty estimation / ensemble model run.

*Please, see our answer to a previous similar comment about the limitations of 1D model. Again, it clearly arises that the Reviewer was misled by our unclear description of the goal of the study, which is not about evaluating the performance of a model, but about how to extract information about cause-effect relationship from a dataset of hydrological variables describing the response to precipitation of the pyroclastic soil mantle of the studied slope. We analyzed the data as if they came from field monitoring, although, to get a richer dataset, we generated a synthetic dataset. The model, already developed, calibrated, and validated in previous studies (Greco et al., 2018), was here used just as a tool to generate the synthetic dataset, by coupling it with the NRSP stochastic model of rainfall. Hence, a sensitivity analysis of the model output to parameters is out of the scope of this study.*

Section 2.2.3/Line 362ff: Referring back to my comment to line 332ff: You mention that you extract variables before the onset of each rainfall event, as the "would be measurable in the field". E.g. those are aquifer water level – which, I assume, is largely different on the top of the slope from the bottom of the slope.

*As we already replied to a previous comment from this Reviewer, the groundwater table depth is indeed variable throughout the slope (observations made in two piezometers, recently installed at two different altitudes along the slope, confirm that the groundwater table depth may be quite different). However, the use that we make of the groundwater level information is to discriminate "low" levels (clusters 1 and 3 of Figures 8, 9 and 10) from "high" levels (cluster 2 of Figures 8, 9 and 10) or "very high" levels (cluster 4 of Fig. 10). Depending on the availability of monitoring instruments, this could be made with a single piezometer, as well as with several piezometers (but, although with different levels, if the groundwater level in a piezometer is high, it will be likely high also in the others, unless they are so far from each other that they are monitoring disconnected groundwater systems). This aspect will be better clarified in the discussion of the results of the revised manuscript.*

Line 370: An actual quantification of soil water content based on satellite observations is hard (rather than a relative value), especially on such small scales – this limitation should be mentioned.

*We will briefly mention the limitations of satellite products compared to field observations.*

Section 2.3.1

Unfortunately, it is hard to understand how you obtain your dataset with triplets of variables. Why did you chose three of the four to predict? You are predicting simulated change in soil water storage, right? What is the time interval? Or is it just aggregated values per rainfall event? How many datapoints? And what are your variable inputs? Only the four mentioned variables? Why did you not also run the 1D physically based model with various parameterizations – I assume the outputs would have looked quite different. Also, why did you chose RF, and how did you decide for number of trees, splits etc (hyperparameters)?

*Again, the Reviewer has been misled by the confusion in the description of the goal of this study. So, we have already replied to the point about evaluating the effects of parameters uncertainty/variability on model output, which is out of the scope of this study.*

*Some of the requested information is already given in the manuscript: we evaluate the change of soil storage between after and before any rainfall event; the number of data is given (around 53000); the choice of the variables is described as the outcome of Random Forest (RF) analysis. We did not test other choices of the variables to be monitored, as we wanted to stick to what can be easily obtained by means of currently available instruments (i.e., satellite products, field soil moisture measurement networks, piezometers, stream water stage sensors).*

*About the choice of the RF, as we already pointed out in a previous answer, it was made to mimic what could be done if, rather than synthetically generated data, one was handling real field monitoring data. In fact, we were mostly looking for a way to identify the major cause-effect relationships between (measurable) inputs and outputs before (possibly, but not necessarily) building a model for the interpretation of such relationships, rather than evaluating the sensitivity of an (already available) model output to variations in the input, and Random Forest allows quantifying the information content of each considered input variable without introducing any mathematical model structure, but just relying on the application of logical operators (IF-THEN-ELSE) between the variables.*

*It looks clear that, aiming at brevity, we gave too little information about how the RF was implemented. Here we provide detailed information to this Reviewer about the training and validation of the RF model, as well as about the choice of hyperparameters.*

*To evaluate the performance of the Random Forest model, the cross-validation technique was used. In cross-validation, the dataset is divided into k equal parts, also known as folds. Then, for each fold, the Random Forest model is trained on the remaining k-1 folds of the data and tested on the remaining fold. We chose k=5, so that the process was repeated 5 times, every time using a different fold (20% of the dataset) as the validation set. A performance metric was calculated for each fold, to estimate how well the RF model perform on new data. We used the explained variance score, computed as follows:*

$$metric = 1 - \frac{Var(y - \hat{y})}{var(y)}$$

$Var(y - \hat{y})$ *and* $Var(y)$ *are the variance of prediction errors and actual values respectively. Higher values of explained variance indicate better performance. In addition, the tuning of the hyperparameter of the model was performed based on the cross-validation results to select the optimal set of hyperparameters. In other words, the random forest model was fitted k-times to the data provided by the cross-validation, changing the value of the following hyperparameters once at time:*

- *n_estimators: the number of trees to build in the forest;*
- *max_depth: the maximum depth of each decision tree in the forest;*
- *min_samples_leaf: the minimum number of samples required to be at a leaf node;*
- *max_features: the number of features to consider when looking for the best split.*

*The following plots show the trend of the hyperparameters vs. the performance metric, the explained variance score. The blue and red lines are the average values of the performance metric computed for the train datasets and the test dataset, respectively, provided by the cross-validation process. The gray and light gray bands represent average values of the metric +/- the standard deviation and two times the standard deviation, respectively.*

[Figure]

*According to the evidence shown by the previous plots, the search of the optimal parameters was carried out using the following ranges for the hyperparameters:*

- *n_estimators was fixed to 10;*
- *max_depth: [5, 10, 15];*
- *min_samples_leaf: [1, 3, 5, 7, 9],*
- *max_features: [2,3].*

*The obtained best parameters are max_depth=10; min_samples_leaf=3; max_features=3.*

*As well known, RF algorithm can work well with high dimensional or multidimensional data, but having a high number of features can lead to overfitting. Therefore, it's important to adjust the hyperparameters (e.g., max_features) to prevent overfitting and create a robust model. Anyway, the synthetic dataset is characterized by only three features (the three variables quantifying antecedent conditions), and a very large number of samples (more than 50000), hence overfitting could be excluded, also owing to the cross-validation method used for model training.*

Section 2.3.2

Again, what data exactly do you cluster? The covariates described in the previous section? Also (line 411) – you do not really use "spatial" data here, do you?

*The clustering is carried out on the triplets that, based on the results of the RF analysis, seems to be the most suitable to describe the effect of antecedent conditions (prior to the onset of each rainfall event) on the attitude of the soil mantle to retain infiltrating rainwater.*

*In line 411 we meant that k-means clustering evaluates the distance between the dots in the space of the variables to which the clustering is applied. Nothing to do with distance in the field. We will rephrase the sentence to make it clearer.*

Section 3.1

Table 3 vs Table 2: Doesn't this shown effect of the normalization of delta(S) to H simply show that there is a quite good linear correlation between H and delta(S) – why then not simply use a linear regression model?

*It is clear that we gave too much emphasis to the analysis of deltaS, so that the Reviewer was misled. The results just show that, obviously, the more it rains, the more soil storage increases, and if one is interested in evaluating the response of the soil mantle to precipitation, one should look at the ratio between deltaS and H. We will remove Table 2 and just briefly explain the choice of deltaS/H.*

Figure 5: The water levels should not be shown on logarithmic scale (that just hides deviations?). Also, make clear that you compare simulated aquifer water levels (ha) with observed stream water levels (hs). Can you try to argue better for your statement "a direct relationship links the water level in the aquifer and the water level in the stream" based on this? As you also mention in section 4, line 632: "substantial agreement between synthetic and experimental data" – this has to be quantified.

*We beg to disagree on the first remark: if plotted along a Cartesian axis, many of the dots would collapse very close to the zero (the "low" water levels), thus hiding the existence of a cluster containing a large number of antecedent conditions.*

*The agreement between synthetic and field data is indeed what we meant to show in Figure 5 and Figure 6. From those figures, you can directly compare the few measured values of soil moisture of the upper 100 cm of the soil profile with the synthetic data. About the synthetic groundwater level data, they are compared with stream water level data. The reasons for this choice are several:*

*- So far, we have measurements of stream water level, while only recently we have installed two piezometers in the epikarst.*

*- The streams are supplied by groundwater coming from the fractured bedrock with very little contribution of overland runoff (less than 1% of the rainfall) only during the most intense rainstorms (it is revealed by the timing of the observed hydrographs in response to rainfall as well as by measurements of electric conductivity of stream water: Marino et al., 2020), so there might be a close relationship linking stream water level and groundwater level.*

*- Installing piezometers in the fractured limestone is a complex operation, owing to the mechanical resistance of the rock, which obliges to the use of powerful drilling machines; we have recently installed two piezometers (July 2020), but one of them could penetrate the limestone only for less than a couple of meters, as the machine that could be carried in that steep part of the slope (a light one) was not able to drill more depth; the second piezometer, which is at the foot of the slope, in a much less steep terrain, penetrates 16 meters below the ground, but there we have found a different kind of soil mantle (not only pyroclastic soil, but also some meters of alluvial deposits), in total more than 10 meters thick; as we had no clue of the degree of interconnection of the fractured system in the limestone, we decided to extend the pervious part of the piezometer (the filter) to almost the entire penetration depth in the limestone (1,5 meters for the first piezometer, 5 meters for the second one), as a shorter filter at the base of the piezometer (as it is usually done) would increase the risk of not intercepting any connected fracture; in this way, there is more chance for water to enter the piezometer, but, as it may enter at any height along the filter and then pond at the base*

*of the piezometer, we cannot convert the water depth that we measure in the piezometer into a groundwater level; during the 2020/2021 hydrologic year we did not measure any water in the piezometers (2020 was a quite dry year), but in December 2021, after a quite rainy autumn (more than 900 mm between September and December), for the first time water appeared in both the piezometers, confirming that the temporary aquifer actually develops in the epikarst during rainy periods; until summer 2022, the piezometric measurements were made irregularly with a freatimeter, but in autumn 2022 we have installed an automatic sensor inside the piezometer on the steep terrain (the first one), and this winter we have observed a slight increase of groundwater level once the cumulated rainfall from September exceeded 800 mm.*

*- Stream water seems to appear and disappear consistently with the groundwater fluctuations, although, so far, we have too few data to demonstrate it; however, measuring stream water level is much easier than groundwater level in the studied context, and it could be an effective surrogate of groundwater level.*

*- The use that we do with the synthetic groundwater level data (that could be done with field data, either of groundwater or of stream water level) is just to discriminate between "high" level and "low" level, as a proxy to identify active subsurface drainage conditions.*

*The colored dots of Figures 5 and 6 also show that the seasonality of the synthetic variables is consistent with that of the observed variables.*

Technical corrections

Some general language editing is necessary

*We will double-check the English language to remove language, syntax and style mistakes.*

Title, and also in the manuscript: "precipitations" cannot be said – maybe replace with "precipitation events" or similar

*Based also on similar remarks made by another Reviewer, the title will be changed to "Understanding hydrologic controls of sloping soil response to precipitation through Machine Learning analysis applied to synthetic data", and the wrong plural "precipitations" will be corrected throughout the entire manuscript.*

---

## Referee Report (RR1)

Dear Authors,

Thanks for answering with such precision the reviewers report and having propose a new version of the article.

**General comments**

In response to the reviewers concerns, you successfully made clearer, in the introduction, that the objective was in fact to identify the most relevant type of field monitoring when dealing with landslide in a sloping environment, and that the models (1D Richards and rainfall generation) were only supporting tools used to enlarge the dataset on which you applied your method (K-mean clustering + random forest).

That being said, some remarks initiated by the reading of the first version remain, especially about the methodology.

Indeed, if the « aim of this study is to identify the major hydrological processes controlling the response of the slope soil mantle […] through suitable measurable variables » (l.99-102), how can you justify the augmentation of your datapool with such a large synthetic dataset (1000 years)? Is it consistent with the practical and operational purpose of the method?

As you mention, lack of data is a very common, not to say a generalized concern: « However, a complete field monitored dataset is not always possible to be analyzed and, when it exists, it is commonly available for short periods, granting a relatively low measurement density » (l.269-272). If data are almost never available in a sufficient amount, the use of a model for synthetic generation is not only supporting your method, but becomes inherent to it.

Therefore, you should explain more precisely the requirements of your methods in terms of data density, and maybe that would shed some light on your choice of generating so much synthetic data. Is 1000 years of data a requirement to ensure statistical consistency? Would a 20 years chronicle (a somehow more realistic perspective) with reinforced data in terms of resolution (ensuring a hourly time step thanks to simulations when needed) be enough? This point is crucial to position your proposed method in an operational and realistic scope.

Stepping a bit further in this intricate issue between need of data and use of models, we come back to the topic of the information already carried by the models: through conceptualization, calibration and sensitivity analysis, a model (especially a physically-based one) informs about the relationship between your input (rainfall, initial or antecedent conditions) and your output (here variation in water storage).

In your case, by applying your method on such a large synthetic dataset (that with no doubt utterly occults the field data), more that directly analysing the relationship between actual rainfalls, antecedent conditions and the underground response (pore pressure in the soil, groundwater level etc.), you statistically sort the relationship between your model input and output.

Therefore, sorry to insist, but I feel obligated to advise a sensitivity analysis. In particular, Sobol indices can be used to untangle the relative influence of a parameter (or initial condition in your case) or their combined influence on a variable of interest, which is specifically what you also aimed to do with machine learning techniques. The advantages of the sensitivity analysis are that it doesn't require much data, you can set the range of values over which you want to explore the sensitivity of the model as you wish (focusing on specific parameters or input), and discriminate different sets to mimic seasonality (benefiting from the statistical clustering analysis, but that can also be set based on less profuse climatic data). The drawbacks are a quite demanding number of runs (with a 1D

model, it shouldn't be an issue) and a complete subjugation to the model, which was not the stated as the initial scope of your paper, I agree.

To summarize and conclude, please make clear the reasons that made you create a such long period of synthetic data. If it was not essential to the method, note that it is somehow contradictory with your operational purpose. If it was indeed essential, you cannot rule out both the bias involved in using models to feed your dataset, and the opportunity to compare your own method with a sensitivity analysis.

**Specific comments**

Figure 1: Thanks for having taken my advice.

L.375-378: « It is important to note that […] would lead to difficult application of the model at less detailed scales such as regional and catchment scales ». If the model is only supporting your method in this particular case, the issue of using it to larger scale wouldn't be an issue worth mentioning in the scope of this paper. Here again, some ambiguity between the objective and the method remains.

L.488-490: The computational effort of less than 2 minutes per run is concerning the RF procedure, am I right? How many runs did you end up simulate considering all the combinations of the variables? What is the duration of a 1D model run to compare (much less than 2 minutes I assume)? This may also argue in favor of a sensitivity analysis.

Figure 5: Thanks for clarifying the signification of the scale. You should mention a reservation about simulated values (whether for groundwater or for river level) below the centimeter/millimeter scale, taking into account that you want to mimic a field monitoring (therefore including limit and uncertainty in the measurement, especially for low values).

---

## Author Response (AR2)

**Reply to referees**

On behalf of all co-authors I would like to thank the referees for spending more time in this extensive revision of our manuscript. In the following text, you could find the point-by-point responses to every individual comment, in which you will find your comments in regular style and our responses in italic.

**Anonymous referee #1**

Dear Authors,

Thanks for answering with such precision the reviewers report and having propose a new version of the article.

*Thank you for spending more time reading our replies and the revised version.*

General comments

In response to the reviewers concerns, you successfully made clearer, in the introduction, that the objective was in fact to identify the most relevant type of field monitoring when dealing with landslide in a sloping environment, and that the models (1D Richards and rainfall generation) were only supporting tools used to enlarge the dataset on which you applied your method (K-mean clustering + random forest).

*Thank you for appreciating the changes to the manuscript, that were motivated by your comments.*

That being said, some remarks initiated by the reading of the first version remain, especially about the methodology.

Indeed, if the « aim of this study is to identify the major hydrological processes controlling the response of the slope soil mantle […] through suitable measurable variables » (l.99-102), how can you justify the augmentation of your datapool with such a large synthetic dataset (1000 years)? Is it consistent with the practical and operational purpose of the method?

As you mention, lack of data is a very common, not to say a generalized concern: « However, a complete field monitored dataset is not always possible to be analyzed and, when it exists, it is commonly available for short periods, granting a relatively low measurement density » (l.269-272). If data are almost never available in a sufficient amount, the use of a model for synthetic generation is not only supporting your method, but becomes inherent to it.

Therefore, you should explain more precisely the requirements of your methods in terms of data density, and maybe that would shed some light on your choice of generating so much synthetic data. Is 1000 years of data a requirement to ensure statistical consistency? Would a 20 years chronicle (a somehow more realistic perspective) with reinforced data in terms of resolution (ensuring a hourly time step thanks to simulations when needed) be enough? This point is crucial to position your proposed method in an operational and realistic scope.

*Thank you for raising this interesting issue. Indeed, long-lasting continuous field monitoring records at hillslopes are rare, and especially as regards critical conditions, such as landslides, they always contain too few data to allow significant statistical analyses. In our case, the clustering analysis of the soil mantle response shows that what we define "effective drainage conditions" (cluster 4 in Figures 10 and 11), quite interesting for understanding the physical processes active in the slope, consist of about 2% of the dataset. Hence, a dataset of only 20 years (containing about 1000 rainfall events) would hardly allow identifying this slope condition. In this respect, increasing the temporal resolution of the dataset would not provide useful information. In fact, the antecedent conditions controlling the response of the soil mantle are linked to*

relatively slow processes (i.e., water accumulation in/drainage from the uppermost meter of the soil mantle, and variations of the water table of the shallow aquifer), which would be captured also with daily resolution. You have been probably misled by our unfortunate word choice « low measurement density », which should be much better written as « small number of measurements » (we have indeed rephrased it in the revised manuscript: lines 274-275). By the way, again referring to the conditions corresponding to cluster 4 (very high groundwater level and uppermost part of the soil mantle wetter than field capacity), they occur after an exceptionally long period of continuous rainfall, which would be unlikely observed in a 20-years long rainfall record. Of course, this does not imply that a dataset of 1000 years at hourly resolution is strictly necessary for our analysis, but ML techniques easily handle big amounts of data, and the simulation of 1000 years with the 1D Richards' equation model had to be carried out only once. Hence, we went for 1000 years, but some hundreds would likely have been enough. To quantify this aspect, we have repeated the ML analyses for shorter durations of the synthetic record.

[Figure]

Figure R1-1. Clustering results of the synthetic data triplets $(\theta_{100}, h_a, \Delta S/H)$ represented in the space $(\theta_{100}, h_a, H)$ for various durations of the synthetic dataset. The right panels show the silhouette score, indicating the optimal number of clusters.

*The above graphs show the spatial arrangements of the data in the space ($\theta_{100}, h_a, H$) for different record durations (i.e., from 10 years to 1000 years), and the corresponding graphs with the values of the silhouette score for different choices of the number of clusters, defined in the space ($\theta_{100}, h_a, \Delta S/H$). Interestingly, the silhouette graphs tend to the same shape for long durations, and at least 50 years are required to obtain that the optimal number of clusters should be four, as for the 1000-year long record. The percentages of the elements belonging to each cluster also become stable for series durations longer than 100 years (e.g., the size of the fourth cluster -very wet conditions- varies between 1.8% and 2.3% of the total number of data considering durations between 100 years and 1000 years; the size of the first cluster -dry conditions- becomes stable around 30% of the data for durations longer than 100 years).*

*Looking at the results of the RF analysis, aiming at identifying the role of the three tested variables (i.e., $\theta_{100}$, $h_a$ and $H$) on the possible prediction of the response of the soil mantle to precipitation, the following graphs show how the estimated importance features of the three variables change, when the RF model is trained with different synthetic record durations (between 20 and 1000 years). For each considered duration, the RF model has been trained ten times with the cross-validation technique described in appendix B. The randomness of the choice of the training and validation sets implies that the estimated importance features are different for each trained RF model. However, if the dataset contains enough information, the estimated importance features of the variables should show only small changes. The two graphs show the changes of the mean value and of the standard deviation of the importance features, estimated from the 10 trained RF models, with different durations of the synthetic dataset. It looks clear that, for record durations longer than about 200 years, the estimated means become stable, and the standard deviations become small.*

[Figure]

*Figure R1-2. Random Forest feature importance for various durations of the synthetic data set: mean value (left panel); standard deviation (right panel).*

*These results seem to indicate that, in the studied case, a record of about 100-200 years could be long enough to convey the information needed to identify the major hydrological controls of the response of the soil mantle. In the revised manuscript, we have added a short sentence to inform the reader about the reasons for the choice of 1000 years (lines 280-285: "The choice of such a long synthetic series has been made to obtain an amount of data, representative also of conditions rarely occurring at the slope, large enough to ensure significance of the analyses carried out with ML techniques. In this respect, it is worth noting that the adopted clustering and Random Forest techniques allow easily handling big amounts of data without unaffordable computational burden.").*

Stepping a bit further in this intricate issue between need of data and use of models, we come back to the topic of the information already carried by the models: through conceptualization, calibration and sensitivity analysis, a model (especially a physically-based one) informs about the relationship between your input (rainfall, initial or antecedent conditions) and your output (here variation in water storage).

In your case, by applying your method on such a large synthetic dataset (that with no doubt utterly occults the field data), more that directly analysing the relationship between actual rainfalls, antecedent conditions and the

underground response (pore pressure in the soil, groundwater level etc.), you statistically sort the relationship between your model input and output.

*We are not sure that we fully understand the Reviewer's point here. The following figure shows the (few) experimental triplets, clustered in the same way as we have made with the synthetic data.*

[Figure]

*Figure R1-3. Clustering results of the measured data triplets ($\theta_{100}, h_a, \Delta S/H$) represented in the space ($\theta_{100}, h_a, H$) for various durations of the synthetic dataset.*

*The results have been plotted in the space ($\theta_{100}, h_a, H$), by replacing the available data of $h_s$ (i.e., stream water level) with the estimated corresponding aquifer level $h_a$ (indeed, we have only recently installed two piezometers in the field, but we do not have available aquifer level data so far). To do this, we used the conceptual relationships, used in the physically based model, to link the aquifer level to the estimated stream discharge and, in turn, to the stream water level. This is only a rescaling of the $h_s$ axis, which does not significantly affect the spatial arrangement of the experimental dots. The total number of field experimental data is too small to draw quantitative interpretations, but the obtained four clusters resemble, to a reasonable extent, those of the synthetic dataset.*

*Looking at what was the response of the soil mantle, in terms of the distribution of $\Delta S/H$ in the four clusters, plotted as boxplots in the following figure, it looks clear that the four clusters, identified from the experimental data, correspond to similar responses as those from the synthetic dataset (plotted in Figure 12 of the paper, here reproduced for your convenience): impeded drainage for cluster 3; strong drainage for cluster 4; similar response for clusters 1 and 2, with "slightly smaller $\Delta S/H$ for cluster 1" (this is what we wrote at lines 624-625 of the previous version of the manuscript, commenting the synthetic responses).*

[Figure]

*Figure R1-4. Distribution of the slope response ΔS/H for the data in each cluster: field data (left panel); synthetic data (right panel).*

*Hence, we don't understand why the Reviewer claims that synthetic dataset, being too numerous, "with no doubt utterly occults the field data". Maybe, here we misunderstand something of the Reviewer's comment, but we believe that the behavior of the synthetic data is a good reproduction of the reality, which does not seem to introduce any biased sorting of the data.*

Therefore, sorry to insist, but I feel obligated to advise a sensitivity analysis. In particular, Sobol indices can be used to untangle the relative influence of a parameter (or initial condition in your case) or their combined influence on a variable of interest, which is specifically what you also aimed to do with machine learning techniques. The advantages of the sensitivity analysis are that it doesn't require much data, you can set the range of values over which you want to explore the sensitivity of the model as you wish (focusing on specific parameters or input), and discriminate different sets to mimic seasonality (benefiting from the statistical clustering analysis, but that can also be set based on less profuse climatic data). The drawbacks are a quite demanding number of runs (with a 1D model, it shouldn't be an issue) and a complete subjugation to the model, which was not the stated as the initial scope of your paper, I agree.

*As you suggest, we have carried out a variance-based sensitivity analysis. Before commenting the obtained results, we would like to stress that there are some other drawbacks of the SA, that you do not mention in your comment.*

*First, differently from when a SA is carried out to evaluate the influence of (the uncertainty of) model parameters on the output, in this case the aim would be to identify the effects of different antecedent conditions on the response of the soil mantle of the slope to precipitation inputs. However, the variables that we use in our ML analysis cannot be either the actual antecedent (initial) condition, nor the actual precipitation input. In fact, for the sake of simplicity (and also to stick to what can be easily measured/handled for practical operational purposes), as antecedent conditions we have considered the mean water content of the uppermost meter of the soil mantle (the initial condition is instead the water content profile throughout the whole mantle) and the water level in the underlying perched aquifer (this latter is indeed an initial condition, but in our model it is physically (and mathematically) linked to the soil moisture at the base of the soil mantle (please, refer to Greco et al. (2018) for details about the equations), so it affects also the initial soil moisture profile).*

*Hence, to carry out the SA, we have been obliged to simplify the initial soil moisture profile in the bi-linear format shown in the following sketch (we already used a similar format in Marino et al. (2021), where you can see to what extent it resembles, at least in many cases, the actual moisture profiles observed in the field):*

[Figure]

*Figure R1-5. Simplified initial moisture profile in the soil mantle, with the corresponding water potential profile, and its relationship with the underlying perched aquifer water level.*

*The rainfall events that are the input of model simulations are also much more complex than the simple total rain depth that we used for our ML analysis of cause-effect relationships. In this case, we carried out the SA considering rainfall events with constant intensity (i.e., rectangular hyetographs), that could be characterized by means of total duration and depth.*

*Summarizing, the SA analysis cannot be easily carried out considering the actual initial and boundary conditions. In our opinion, this is a point that moves the needle in favor of the Random Forest analysis.*

*The second drawback is related to the generation of the set of combinations of input parameters, required to carry out the variance-based SA. In fact, to avoid introducing a bias in the estimated output variance, the set of generated input values should be consistent with their probability distributions, which cannot be simplistically assumed Normal (as it is usually made when the SA aims at quantifying the effects of model parameter uncertainty). In the case of $\theta_{100}$ and $h_a$, we have little information from our field data (about 50 couples, as can be seen in figure 5 of the manuscript, here reproduced for your convenience):*

[Figure]

*Figure R1-6. Field monitored mean volumetric water content in the upper meter of the soil profile ($\theta_{100}$) and water depth in the Castello stream ($h_s$) compared with simulated data (the vertical axes are plotted in logarithmic scales to help visualizing small water levels).*

Based on the small set of experimental data on antecedent conditions, it is possible to roughly visualize the frequency distributions of the observed values of $\theta_{100}$ and $h_a$, here compared with those of $\theta_{100}$ and $h_s$ of the synthetic dataset (reproduced from Fig. 7 of the manuscript).

[Figure]

*Figure R1-7. Frequency distributions of (a) synthetic and (b) observed initial moisture of the uppermost meter of the soil mantle, $\theta_{100}$; frequency distribution of (c) synthetic antecedent water depth in the Castello stream, $h_s$, and (d) observed perched aquifer water level, $h_a$.*

The few observed data (by the way, somehow confirming the shape of the distributions obtained with the model chain used for the generation of the synthetic dataset) indicate that the distributions of the antecedent values are neither Normal nor uniform.

About the precipitation input, the available record of 17 years, which had been used for the calibration of the NRSP model (as described in Appendix A), contains enough information about the distributions of the values of rainfall event duration and depth (see figure A4 of appendix A, here reproduced for your convenience).

[Figure]

*Figure R1-8. Scatterplot of total rainfall event depth (H) vs. rainfall event duration (D). The events have been sorted within the rainfall datasets by considering a separation "dry" interval of 24 hours with less than 2 mm rainfall. The blue dots represent events extracted from the 17 years experimental rainfall dataset, while the grey dots represent events extracted from the 1000 years synthetic rainfall dataset.*

*The third drawback, also regarding the generation of the set of combinations of the variables $(\theta_{100}, h_a, H)$, is related to the issue of the existence of correlation between the input variables. For instance, event rainfall depth H and duration D clearly exhibit significant correlation (see figure R1-8), but also $\theta_{100}$ and $h_a$, both related to previous precipitation history, may show some degree of correlation, which should be considered when the combination of input variables are generated, in order to carry out rigorously the variance-based sensitivity analysis.*

*All these considerations have been made here just to underline that carrying out a SA with the purpose of characterizing the input-output relationships in a system is not an easy task, which implies assumptions and issues which make the interpretation of the results far from being obvious.*

*This said, the SA has been carried out, based on the methodology outlined by Sobol (2001), which is implemented in the Sensitivity Analysis Library in Python - SALib toolbox (Herman and Usher, 2017; Iwanaga et al., 2022). To reduce the error rates in the calculation of the sensitivity index, the Saltelli's sampling scheme is used to generate the uniform sample set (Saltelli, 2002; Saltelli et al., 2010). Specifically, $N \times (2P + 2) = 65536$ triplets of $(\theta_{100}, h_a, H)$ have been generated, with N=8192 (a power of 2) and P=3 the number of inputs, and each time the physically based model has been run to obtain the corresponding soil mantle response $\Delta S/H$. For each sampled value of H, the corresponding event duration D has been obtained by means of a power-law fitting relationship of the experimental rain events plotted in figure R1-8 ($D = 0.282 \times H^{1.525}$). The calculated sensitivity indices are given in the following table.*

| Variable | $S_{tot}$ | $S_1$ (single parameter variations) | $S_2$ (mutual interactions) | |
|---|---|---|---|---|
| $\theta_{100}$ | 0.532 | 0.471 | $(\theta_{100}, h_a)$ | 0.002 |
| $h_a$ | 0.058 | 0.058 | $(\theta_{100}, H)$ | 0.060 |
| $H$ | 0.469 | 0.412 | $(h_a, H)$ | 0.000 |

*The obtained results shed more light on the different meaning of a variance-based SA, compared to the importance feature estimated with the RF model. The SA explains how the variability of the output (here $\Delta S/H$) is related to the variability of the inputs, and it looks clear that the variability of $\theta_{100}$ and H strongly (almost completely) affects the variability of $\Delta S/H$. Differently, the RF analysis aimed at estimating what are the most informative variables, useful to make good predictions of the output. While the SA only looks at the variation of the output, the RF also looks at how well the output is predicted, compared to "real" observations (in this case, synthetic). We have added some paragraphs in section 2.3.1 to highlight the difference between RF*

*importance analysis and SA (lines 435-438; lines 443-452; lines 468-474), and the comparison of the results of SA with RF (lines 546-567, and Table 3).*

*In this respect, while it is physically obvious that, the more variable are the rain and the initial wetness of the soil (i.e., H and $\theta_{100}$), the more variable is the stored precipitation $\Delta S/H$ (i.e., more rain, more storage; dryer soil, more storage possible), the results of the RF analysis are less trivial. The aquifer water level $h_a$, not affecting so much the variability of $\Delta S/H$, as indicated by the SA, is anyway an extremely informative variable, as it allows separating the initial conditions in two families: low levels (clusters 1 and 3), high levels (clusters 2 and 4). In conjunction with the information about rainfall amount and initial soil moisture, this separation strongly improves the capability of predicting the response of the soil mantle to precipitation in terms of $\Delta S/H$ (e.g., the different response of cluster 2 and cluster 3, that share the same values of antecedent $\theta_{100}$, but exhibit a quite different attitude to retain infiltrating rainwater).*

To summarize and conclude, please make clear the reasons that made you create a such long period of synthetic data. If it was not essential to the method, note that it is somehow contradictory with your operational purpose. If it was indeed essential, you cannot rule out both the bias involved in using models to feed your dataset, and the opportunity to compare your own method with a sensitivity analysis.

*We believe that the results and considerations, developed in response to the issues raised by this Referee, have satisfactorily clarified the doubtful points, and have allowed improving the quality of the manuscript.*

Specific comments

Figure 1: Thanks for having taken my advice.

*Indeed, it was a good advice, and the Figure has been improved.*

L.375-378: « It is important to note that […] would lead to difficult application of the model at less detailed scales such as regional and catchment scales ». If the model is only supporting your method in this particular case, the issue of using it to larger scale wouldn't be an issue worth mentioning in the scope of this paper. Here again, some ambiguity between the objective and the method remains.

*You are right, the paragraph between lines 376 and 381 is truly unclear. The model aims at representing a geomorphological setting, air-fall pyroclastic deposits overlying calcareous bedrock resting on steeply inclined slopes, which is found in large areas of Campania (southern Italy), and not only in the Partenio Massif, where the studied slopes belong to. The following figure, adapted from the cited paper by Cascini et al. (2008), sketches the areas where slopes share such characteristics.*

[Figure]

*Figure R1-5 (adapted from Cascini et al., 2008). Air-fall pyroclastic deposits in the Campania region: 1) carbonate bedrock; 2) tuff and lava deposits; 3) flysch and terrigenous bedrock; 4) alluvial and continental deposits; 5) volcanic complexes; 6) isopachous lines of the pyroclastic products from the main eruptions (in brackets eruption data).*

*The slopes of Sarno mounts, Picentini mounts, Lattari mounts, as well as those of Partenio Massif, present pyroclastic mantles with similar physical and geometric characteristics, laying upon bedrock with similar geological features, covered with similar vegetation, and they share similar climate. Consequently, all the slopes of this whole large area (i.e., few thousands of square kilometers) are frequently subjected to rainfall-induced shallow landslides, triggered by similar rain events in similar antecedent conditions (e.g., Di Crescenzo and Santo, 2005; Cascini et al., 2008; Greco et al., 2021). We have modified the Introduction to better highlight that the study area is representative of a geomorphological context quite common in the region region (lines 74-81:" This research focuses on a case study of a slope located in Campania (southern Italy), representative of a wide area frequently hit by destructive rainfall-triggered shallow landslides (e.g., Fiorillo et al., 2001; Revellino et al., 2013). In fact, such geohazards are recurrent along the carbonate slopes covered with unsaturated air-fall pyroclastic deposits, diffuse over an area of few thousand square kilometres around the two major volcanic complexes of the region, the Somma-Vesuvius and the Phlaegrean Fields (Di Crescenzo and Santo, 2005; Cascini et al., 2008)"), and that the aim of the study refers to slopes of the area with characteristics similar as the modelled one (lines 102-105).*

*Obviously, every slope in the area has its own specific features, but the hydrological processes controlling the response of the soil mantle to precipitation events can be considered similar over wide areas, as they are related to large scale (in time and space) processes such as long-term cumulated rainfall and evapotranspiration, and perched aquifer recharge. In this sense, we believe that our simplified 1D slope model (with a homogeneous soil mantle of constant thickness, constant slope inclination, and homogeneous epikarst) could be useful to approximately assess the antecedent conditions that control the response of the soil mantle to precipitation in a wide area. This is the meaning of the fateful expression at lines 376-379 (and also at lines 379-381). To make them clearer, we have modified the paragraph, which now reads (lines 389-399): "The model assumes a homogeneous soil profile and a simplified slope geometry, and indeed it is not aimed at reproducing the details of local flow processes through the unsaturated soil mantle. Consequently, the hydraulic properties of the homogeneous soil layer should be considered as effective properties, useful to reproduce the major features of the infiltration and drainage phenomena. The model is rather used to assess how large-scale (in time and space) hydrological processes, such as long-term cumulated rainfall and evapotranspiration and perched aquifer recharge, control the conditions that affect the response of the soil mantle to precipitation events. In this sense, the obtained results can be considered representative for large areas that share the major geomorphological features of the slopes of Partenio Massif".*

L.488-490: The computational effort of less than 2 minutes per run is concerning the RF procedure, am I right? How many runs did you end up simulate considering all the combinations of the variables? What is the duration of a 1D model run to compare (much less than 2 minutes I assume)? This may also argue in favor of a sensitivity analysis.

*The calibration of the hyperparameters of the four tested RF models took in total less than one hour, leading to the choice of the most informative triplet of input variables $(\theta_{100}, h_a, H)$, as well as to the scores measuring the contribution of each variable. To have an idea of the computational effort of the ML procedure, this time should be summed to the time needed to carry out the simulation of 1000 years (about $8.76 \times 10^6$ hours). Such time is longer than the time required to carry out the 1D model simulations of the SA: the required 65536 rainfall events have variable durations, ranging between 1 hour and few hundreds of hours (see figure 4, above). Considering that the mean rain event duration is about 26.5 hours, the total simulated time results about $1.7 \times 10^6$ hours, indeed shorter than the $8.76 \times 10^6$ hours, corresponding to 1000 years. However, running 65536 separated simulations, instead of a single long model run, implies additional computational time, thus reducing the difference of computational burden (in fact, the simulations for the sensitivity analysis lasted about 10 hours, while the 1000 years single model run took about 28 hours on the same computer). So, although the SA allows saving some computational time, the gain does not look so significant to be the reason to guide the choice of the method of analysis.*

Figure 5: Thanks for clarifying the signification of the scale. You should mention a reservation about simulated values (whether for groundwater or for river level) below the centimeter/millimeter scale, taking into account that you want to mimic a field monitoring (therefore including limit and uncertainty in the measurement, especially for low values).

*Thank you for the suggestion. We have modified the caption of Figure 5, which now reads "Field monitored mean volumetric water content in the upper meter of the soil profile ($\theta_{100}$) and water depth in the Castello stream ($h_s$), compared with synthetic data of $\theta_{100}$ and aquifer water level ($h_a$) (on the vertical axis, plotted in logarithmic scale to help visualizing small water levels and thus not allowing to represent zeros, the values of $h_s$ smaller than the sensitivity of the water level sensor have been plotted as 1 mm; also the smallest simulated values of $h_a$ should be considered equivalent to zero, owing to the limits of any measurement device which could be used for operational field monitoring)".*

*References*

*Cascini, L., Cuomo, S., and Guida, D.: Typical source areas of May 1998 flow-like mass movements in the Campania region, Southern Italy, Eng. Geol., 96, 107-125, https://doi.org/10.1016/j.enggeo.2007.10.003, 2008.*

*Di Crescenzo, G., and Santo, A.: Debris slides–rapid earth flows in the carbonate massifs of the Campania region (Southern Italy): morphological and morphometric data for evaluating triggering susceptibility, Geomorphology, 66(1-4), 255-276, https://doi.org/10.1016/j.geomorph.2004.09.015, 2005.*

*Greco, R., Comegna, L., Damiano, E., Marino, P., Olivares, L., and Santonastaso, G. F.: Recurrent rainfall-induced landslides on the slopes with pyroclastic cover of Partenio Mountains (Campania, Italy): Comparison of 1999 and 2019 events, Eng. Geol., 288, 106160, https://doi.org/10.1016/j.enggeo.2021.106160, 2021.*

*Greco, R., Marino, P., Santonastaso, G. F., and Damiano, E.: Interaction between Perched Epikarst Aquifer and Unsaturated Soil Cover in the Initiation of Shallow Landslides in Pyroclastic Soils, Water, 10, 948, https://doi.org/10.3390/w10070948, 2018.*

*Herman, J., and Usher, W.: SALib: an open-source Python library for Sensitivity Analysis. The Journal of Open Source Software, 2(9), 97, https://doi.org/10.21105/joss.00097, 2017.*

Iwanaga, T., Usher, W., and Herman, J.: Toward SALib 2.0: Advancing the accessibility and interpretability of global sensitivity analyses. Socio-Environmental Systems Modelling, 4, 18155–18155, https://doi.org/10.18174/SESMO.18155, 2022.

Marino, P., Santonastaso, G. F., Fan, X., and Greco, R.: Prediction of shallow landslides in pyroclastic-covered slopes by coupled modeling of unsaturated and saturated groundwater flow, Landslides, https://doi.org/10.1007/s10346-020-01484-6, 2021.

Saltelli, A.: Making best use of model evaluations to compute sensitivity indices. Computer Physics Communications, 145(2), 280–297. https://doi.org/10.1016/S0010-4655(02)00280-1, 2002.

Saltelli, A., Annoni, P., Azzini, I., Campolongo, F., Ratto, M., and Tarantola, S.: Variance based sensitivity analysis of model output. Design and estimator for the total sensitivity index. Computer Physics Communications, 181(2), 259–270. https://doi.org/10.1016/J.CPC.2009.09.018, 2010.

Sobol, I. M.: Global sensitivity indices for nonlinear mathematical models and their Monte Carlo estimates. Mathematics and Computers in Simulation, 55(1–3), 271–280. https://doi.org/10.1016/S0378-4754(00)00270-6, 2001.

**Anonymous referee #3**

Thanks for the revised manuscript. The authors performed major revisions, addressing many of the raised concerns by the other reviewers and myself. The manuscript now clearly is easier to understand, and its objectives better defined, especially for the typical audience of HESS. The appendices explaining the rainfall generator and details of RF setup also help.

*Thank you for acknowledging the improvement of the manuscript.*

However, there is one central point that remains open from my side, concerning my questions about the use of a sensitivity analysis (or uncertainty estimation) of the 1D model, instead of the analysis of its results using RF (similar points also were raised by the other two reviewers):

It is a central assumption in your manuscript that this single realization of a 1D model is an adequate representation of reality to base your paper on an ML-based analysis of its outputs. This is so central to your paper, that I think it still deserves more attention.

I do understand the authors' argument that their study is not an evaluation of the sensitivity or uncertainty of the 1D model. And maybe I should have been more precise with my concern in the first review ("[…]You set up a single model (with a single parameter set) – ignoring the heterogeneity of soil thickness, hydraulic conductivites, etc. one would find along the slope? Given such a simple 1D model, I would at least recommend to perform some kind of sensitivity analysis / parameter uncertainty estimation / ensemble model run."). I try to reformulate: You use one single realization of a calibrated 1D model, with a synthetic rainfall dataset, to understand how seasonal conditions may affect triggering of landslides. Your central assumption here is that the model-based synthetic dataset is an adequate representation of reality. Despite using a calibrated model, I would still assume that there is some uncertainty or equifinality in the model? I.e. do you know how well defined are the decisive parameters? Could you not find various parameter combinations that give almost equally well performance of your 1D model? Potentially even quite different parameter combinations? (not even touching on questions of model structure…) Models with different parameters sets could also exhibit quite different physical behaviour/cause-effect relationships, despite similar performance in terms of calibration data. They should be considered an equally adequate basis for your subsequent RF analysis – and might yield in quite different final results and conclusions. This aspect I still am missing – I would really love to see this addressed (how well defined are the parameters of the 1D model? How much equifinality is there?), or at least it has to be discussed as a shortcoming.

*What this Reviewer is asking about Sensitivity Analysis is something different from what is asked by the other Reviewer. In fact, here the Reviewer is concerned with the effects of the uncertainty of model parametrization on the cause-effect relationships that we have identified in the synthetic data, and thus he suggests going back to the calibration/validation of the 1D Richards' equation model coupled with the underlying perched aquifer hosted in the epikarst, which had been made long time ago (Greco et al., 2013). Most of the parameters of the model were assigned based either on literature indications or on available measured values. The calibration of the remaining parameters (i.e., those of unsaturated soil hydraulic characteristic functions) was carried out based on field measurements of soil water potential and moisture. The parameters, searched with a Genetic Algorithm, were constrained so to ensure the corresponding hydraulic functions to resemble available measurements of water retention and unsaturated hydraulic conductivity, obtained both in the field and in the laboratory (we have added this information at lines 369-373 in the revised manuscript). The figure below, reproduced from Greco et al. (2013), shows an example of the obtained agreement between simulations and measurements.*

[Figure]

*Figure R2-1 (adapted from Greco et al., 2013). Comparison between simulated and observed capillary pressure head at various depths during the period considered for model calibration: slope with leafless trees and low-developed underbrush (above); slope during the vegetation flourishing period (below).*

*A 2D version of the model, with the same structure about the coupling between unsaturated soil cover and perched aquifer, was then successfully applied to the assessment of the multi-year water balance of the slope of Cervinara (Greco et al., 2018). It also allowed reliably interpreting the response of the soil mantle during intense rainstorms, which might result or not in the triggering of landslides (Greco et al., 2021; Marino et al., 2021).*

*Re-evaluating the calibration of the model, carrying out a SA of the output to model parameter uncertainty, would imply introducing a completely different chapter in the manuscript, which is far beyond the scope of this work.*

*In fact, the simplified model (with a single homogeneous soil layer with effective parameters and a simplified slope geometry, with constant inclination and soil mantle thickness, and homogeneous epikarst) aims at representing a geomorphological setting, air-fall pyroclastic deposits overlying calcareous bedrock resting on steeply inclined slopes, which is found in large areas of Campania (southern Italy), and not only in the Partenio Massif, where the studied slopes belong to. The following figure, adapted from the cited paper by Cascini et al. (2008), sketches the areas where slopes share such characteristics.*

[Figure]

*Figure R2-2 (adapted from Cascini et al., 2008). Air-fall pyroclastic deposits in the Campania region: 1) carbonate bedrock; 2) tuff and lava deposits; 3) flysch and terrigenous bedrock; 4) alluvial and continental deposits; 5) volcanic complexes; 6) isopachous lines of the pyroclastic products from the main eruptions (in brackets eruption data).*

*The slopes of Sarno mounts, Picentini mounts, Lattari mounts, as well as those of Partenio Massif, present pyroclastic mantles with similar physical and geometric characteristics, laying upon bedrock with similar geological features, covered with similar vegetation, and they share similar climate. Consequently, all the slopes of this whole large area (i.e., few thousands of square kilometers) are frequently subjected to rainfall-induced shallow landslides, triggered by similar rain events in similar antecedent conditions (e.g., Di Crescenzo and Santo, 2005; Cascini et al., 2008; Greco et al., 2021). We have modified the Introduction to better highlight that the study area is representative of a geomorphological context quite common in the region (lines 74-81:" This research focuses on a case study of a slope located in Campania (southern Italy), representative of a wide area frequently hit by destructive rainfall-triggered shallow landslides (e.g., Fiorillo et al., 2001; Revellino et al., 2013). In fact, such geohazards are recurrent along the carbonate slopes covered with unsaturated air-fall pyroclastic deposits, diffuse over an area of few thousand square kilometres around the two major volcanic complexes of the region, the Somma-Vesuvius and the Phlaegrean Fields (Di Crescenzo and Santo, 2005; Cascini et al., 2008)"), and that the aim of the study refers to slopes of the area with characteristics similar as the (simplified) modelled one (lines 102-105).*

*Obviously, every slope in the area has its own specific features, but the hydrological processes controlling the response of the soil mantle to precipitation events can be considered similar over wide areas, as they are related to large scale (in time and space) processes such as long-term cumulated rainfall and evapotranspiration, and perched aquifer recharge. In this sense, we believe that our simplified 1D slope model (with a homogeneous soil mantle of constant thickness, constant slope inclination, and homogeneous epikarst) could be useful to approximately assess the antecedent conditions that control the response of the soil mantle to precipitation in a wide area. This is the meaning of the (unclear, we agree) expressions at lines 376-381. To make them clearer, we have modified the paragraph, which now reads (lines 389-399): "The model assumes a homogeneous soil profile and a simplified slope geometry, and indeed it is not aimed at simulating the details of local flow processes through the unsaturated soil mantle. Consequently, the hydraulic properties of the homogeneous soil layer should be considered as effective properties, useful to reproduce the major features of the infiltration and drainage phenomena. The model is rather used to assess how large-scale (in time and space) hydrological processes, such as long-term cumulated rainfall and evapotranspiration, and perched aquifer recharge, control the conditions that affect the response of the soil mantle to precipitation events. In this sense, the obtained results can be considered representative for large areas that share the major geomorphological features of the slopes of Partenio Massif".*

*This said, we believe that it is not worth carrying out also a SA of the output to the uncertainty of model parameters. As suggested, we have added a "warning" about the limitations of the obtained results at the beginning of section 3 (Results and discussion) (lines 506-513: "The analysis of the physical behavior of the*

*studied slopes is based on the results of model simulations, as if they satisfactorily resemble what could be measured in the field. Indeed, the uncertainty of model parameters may affect the identified cause-effect relationships. However, during the calibration of model, field measurements of the hydraulic behavior of the involved soil were considered (Greco et al., 2013), thus the major features of the hydrological processes occurring in the slope are considered reliably reproduced in the synthetic dataset").*

*References*

*Cascini, L., Cuomo, S., and Guida, D.: Typical source areas of May 1998 flow-like mass movements in the Campania region, Southern Italy, Eng. Geol., 96, 107-125, https://doi.org/10.1016/j.enggeo.2007.10.003, 2008.*

*Di Crescenzo, G., and Santo, A.: Debris slides–rapid earth flows in the carbonate massifs of the Campania region (Southern Italy): morphological and morphometric data for evaluating triggering susceptibility, Geomorphology, 66(1-4), 255-276, https://doi.org/10.1016/j.geomorph.2004.09.015, 2005.*

*Greco, R., Comegna, L., Damiano, E., Guida, A., Olivares, L., and Picarelli, L.: Hydrological modelling of a slope covered with shallow pyroclastic deposits from field monitoring data, Hydrol. Earth. Syst. Sci., 17, 4001–4013, https://doi.org/10.5194/hess-17-4001-2013, 2013.*

*Greco, R., Comegna, L., Damiano, E., Marino, P., Olivares, L., and Santonastaso, G. F.: Recurrent rainfall-induced landslides on the slopes with pyroclastic cover of Partenio Mountains (Campania, Italy): Comparison of 1999 and 2019 events, Eng. Geol., 288, 106160, https://doi.org/10.1016/j.enggeo.2021.106160, 2021.*

*Greco, R., Marino, P., Santonastaso, G. F., and Damiano, E.: Interaction between Perched Epikarst Aquifer and Unsaturated Soil Cover in the Initiation of Shallow Landslides in Pyroclastic Soils, Water, 10, 948, https://doi.org/10.3390/w10070948, 2018.*

*Marino, P., Santonastaso, G. F., Fan, X., and Greco, R.: Prediction of shallow landslides in pyroclastic-covered slopes by coupled modeling of unsaturated and saturated groundwater flow, Landslides, https://doi.org/10.1007/s10346-020-01484-6, 2021.*